# SON and SRRM2 are essential for nuclear speckle formation

İbrahim Avşar Ilik[1], Michal Malszycki[1,2†], Anna Katharina Lübke[1,2†], Claudia Schade[1], David Meierhofer[1], Tuğçe Aktaş[1]*

[1]Max Planck Institute for Molecular Genetics, Berlin, Germany; [2]Freie Universität Berlin, Berlin, Germany

**Abstract** Nuclear speckles (NS) are among the most prominent biomolecular condensates. Despite their prevalence, research on the function of NS is virtually restricted to colocalization analyses, since an organizing core, without which NS cannot form, remains unidentified. The monoclonal antibody SC35, raised against a spliceosomal extract, is frequently used to mark NS. Unexpectedly, we found that this antibody was mischaracterized and the main target of SC35 mAb is SRRM2, a spliceosome-associated protein that sharply localizes to NS. Here we show that, the core of NS is likely formed by SON and SRRM2, since depletion of SON leads only to a partial disassembly of NS, while co-depletion of SON and SRRM2 or depletion of SON in a cell-line where intrinsically disordered regions (IDRs) of SRRM2 are genetically deleted, leads to a near-complete dissolution of NS. This work, therefore, paves the way to study the role of NS under diverse physiological and stress conditions.

*For correspondence:
aktas@molgen.mpg.de

†These authors contributed equally to this work

Competing interests: The authors declare that no competing interests exist.

## Introduction

Nuclear speckles (NS) are membraneless nuclear bodies (*Banani et al., 2017*; *Shin and Brangwynne, 2017*) in the interchromatin-space of the nucleus that contain high concentrations of RNA-processing and some transcription factors but are devoid of DNA (*Spector and Lamond, 2011*). Under normal conditions, they appear as irregularly shaped, dynamic structures that show hallmarks of phase-separated condensates, such as fusion and deformation under pressure in living cells (*Chen and Belmont, 2019*; *Zhang et al., 2016*). Despite their prevalence, the function of NS remains largely unknown, although they have been proposed to act as reservoirs for splicing factors, and association with NS have been shown to correlate with enhanced transcription and RNA-processing (*Chen and Belmont, 2019*; *Galganski et al., 2017*). NS have been shown to be involved in replication of herpes simplex virus (*Chang et al., 2011*), processing and trafficking of Influenza A virus mRNA (*Mor et al., 2016*), detaining repetitive RNA originating from the transcription of repeat expanded loci that trigger Huntington's disease, spinocerebellar ataxia and dentatorubral–pallido-luysian atrophy (*Urbanek et al., 2016*), but also repetitive RNA from artificial constructs that produce RNA capable of phase-separation in vitro (*Jain and Vale, 2017*). Studying the role of NS involves visualizing them with a fluorescently tagged factor that localizes to NS, or the use of antibodies that show specific staining of NS. Similar to nucleoli and other membraneless bodies of the nucleus, NS disassemble during early stages of mitosis, and assemble back following telophase (*Galganski et al., 2017*). Several protein kinases are thought to be involved in this process, such as DYRK3, chemical inhibition of which leads to aberrant phase-separation (*Rai et al., 2018*). Overexpression of DYRK3, or CLK1 on the other hand leads to dissolution of NS in interphase cells, underscoring the importance of phosphorylation in NS integrity (*Rai et al., 2018*; *Sacco-Bubulya and Spector, 2002*). Unlike several other biomolecular condensates, a specific core necessary for NS formation has not yet been identified, and it has been hypothesized that stochastic self-assembly of NS-associated factors could lead to the formation of NS (*Dundr and Misteli, 2010*; *Spector and*

**eLife digest** Most cells store their genetic material inside a compartment called the nucleus, which helps to separate DNA from other molecules in the cell. Inside the nucleus, DNA is tightly packed together with proteins that can read the cell's genetic code and convert into the RNA molecules needed to build proteins. However, the contents of the nucleus are not randomly arranged, and these proteins are often clustered into specialized areas where they perform their designated roles.

One of the first nuclear territories to be identified were granular looking structures named Nuclear Speckles (or NS for short), which are thought to help process RNA before it leaves the nucleus. Structures like NS often contain a number of different factors held together by a core group of proteins known as a scaffold. Although NS were discovered over a century ago, little is known about their scaffold proteins, making it difficult to understand the precise role of these speckles.

Typically, researchers visualize NS using a substance called SC35 which targets specific sites in these structures. However, it was unclear which parts of the NS this marker binds to. To answer this question, Ilik et al. studied NS in human cells grown in the lab. The analysis revealed that SC35 attaches to certain parts of a large, flexible protein called SRRM2. Ilik et al. discovered that although the structure and sequence of SRMM2 varies between different animal species, a small region of this protein remained unchanged throughout evolution.

Studying the evolutionary history of SRRM2 led to the identification of another protein with similar properties called SON. Ilik et al. found that depleting SON and SRRM2 from human cells caused other proteins associated with the NS to diffuse away from their territories and become dispersed within the nucleus. This suggests that SRMM2 and SON make up the scaffold that holds the proteins in NS together.

Nuclear speckles have been associated with certain viral infections, and seem to help prevent the onset of diseases such as Huntington's and spinocerebellar ataxia. These newly discovered core proteins could therefore further our understanding of the role NS play in disease.

*Lamond, 2011*; *Tripathi et al., 2012*). One of the most frequently used reagents to locate NS is the monoclonal antibody SC35, which was raised against biochemically purified spliceosomes (*Fu and Maniatis, 1990*), and reported to be an antibody against SRSF2 (*Fu and Maniatis, 1992*). Testament to the importance of this antibody, NS are also referred to as 'SC35 domains'. Although, the name SC35 and SRSF2 are used synonymously and to annotate orthologues of SRSF2 not only in mammalian species but also in species such as *D. melanogaster* and *A. thaliana*, mAb SC35 is reported to cross-react with SRSF1, and potentially with other SR-proteins as well (*Fu et al., 1992*; *Neugebauer and Roth, 1997*). Furthermore, fluorescently tagged SRSF2 shows staining patterns incompatible with mAb SC35 stainings under identical experimental conditions (*Politz et al., 2006*; *Sakashita and Endo, 2010*; *Sharma et al., 2010*; *Tripathi and Parnaik, 2008*). Intrigued by these inconsistencies, we undertook a systemic re-characterization of the mAb SC35 and its cellular targets.

## Results

### IP-MS reveals endogenous targets of mAb SC35

In order to characterize the cellular targets of the SC35 mAb, we carried out an Immunoprecipitation Mass-Spectrometry (IP-MS) experiment. Whole-cell extracts prepared from HAP1 cells were used to immunoprecipitate endogenous targets SC35 mAb, with a matched IgG mAb serving as a control. The immunoprecipitated proteins were then analyzed by mass-spectrometry (see Methods for details). In total, we identified 432 proteins that were significantly enriched in the SC35 purifications compared to controls (p<0.05, at least two peptides detected in each biological replicate). Surprisingly, the most enriched protein in the dataset, both in terms of unique peptides detected, total intensities and MS/MS spectra analyzed, is neither SRSF2 nor one of the canonical SR-proteins (*Manley and Krainer, 2010*), but a high-molecular weight RNA-binding protein called SRRM2 (*Figure 1A*, *Figure 1—figure supplement 1A*), an NS-associated protein with multiple RS-repeats

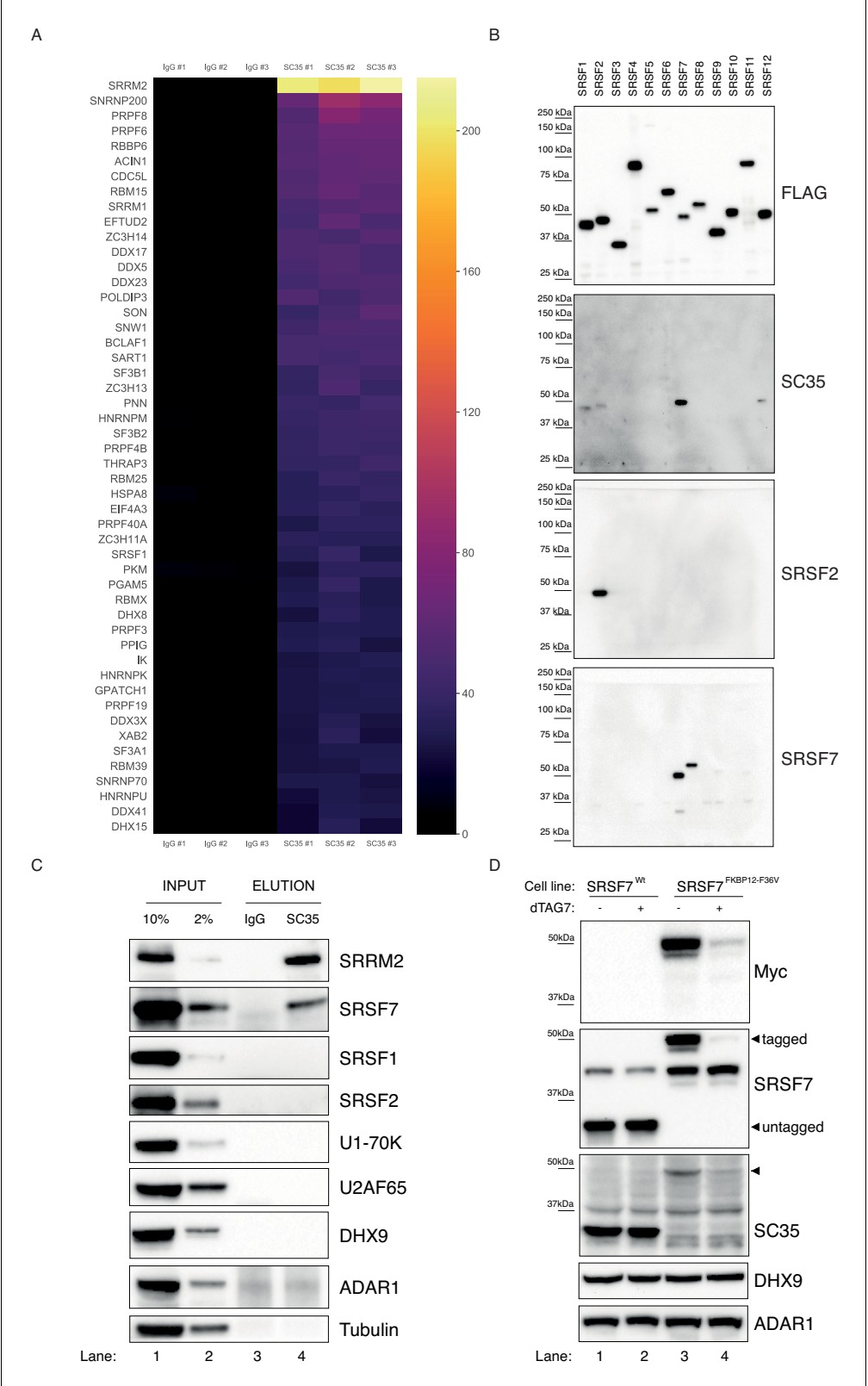

**Figure 1.** SC35 mAb immunoprecipitation followed by MS identifies SRRM2 as the top hit. (**A**) The Top50 hits identified by the MS are depicted on a heatmap showing the number of unique peptides detected for each protein. Also see *Figure 1—figure supplement 1A* for an intensity vs MS/MS spectra plot and *Supplementary file 2* for raw MaxQuant results (**B**) Streptavidin pull-down of biotin tagged ectopically expressed SRSF proteins 1 to 12 are blotted with FLAG antibody in order to show the amounts of loaded proteins on PAGE. Western blot using mAb SC35 detects SRSF7 with

*Figure 1 continued on next page*

*Figure 1 continued*

highest sensitivity in comparison to all other SRSF proteins, but also weakly reacts with SRSF1, 2, and 12. Specific antibodies against SRSF2 and SRSF7 are used to validate the authenticity of the purified proteins from stable cell lines in corresponding lanes and blots. (C) SC35-IP performed on lysates from wild-type HEK293 cells identifies SRRM2 as the most enriched protein with a weaker enrichment for SRSF7 but no enrichment for SRSF1 or SRSF2 using western blotting (compare Lane 4 across blots). (D) Homozygous knock-in of the 2xMyc-FKBP12$^{F36V}$ tag into *SRSF7* gene locus shifts the SC35 band from 35 kDa to 50 kDa (compare Lanes 1 and 3 on the SC35 blot) and upon induction of degradation with dTAG7 the shifted band is lost (compare Lanes 3 and 4 in Myc, SRSF7, and SC35 blots). This blot validates that the 35 kDa band identified by mAb SC35 blots corresponds to SRSF7. The online version of this article includes the following figure supplement(s) for figure 1:

**Figure supplement 1.** SC35 pull-down followed by MS identifies SRRM2 as the top hit and multiple spliceosomal components are co-purified together with SRRM2.
**Figure supplement 2.** A validation for all stable cell lines with transgenic SRSF proteins show only weak localization to the NS.

(*Blencowe et al., 2000*). Analysis of the top 108 targets, corresponding to the third quartile, using the STRING database (*Szklarczyk et al., 2019*) shows a clear enrichment for the spliceosome and NS (*Figure 1—figure supplement 1B*), validating the experimental approach. We were also able to detect all SR-proteins in our dataset, however their scores are dwarfed by that of SRRM2's (*Figure 1—figure supplement 1C*). Thus, contrary to initial expectations, the IP-MS results strongly suggest but do not prove that SC35 mAb primarily recognizes SRRM2, at least in the context of an immunoprecipitation experiment.

## SRSF7 is the 35 kDa protein recognized via western blots with mAb SC35

Before exploring SRRM2 as a potential mAb SC35 target protein, we decided to first take an unbiased look at SR-proteins and the ability of mAb SC35 to recognize them. To this end, we cloned all 12 canonical SR-proteins in humans (*Manley and Krainer, 2010*) into an expression plasmid, and created stable-cell lines expressing these proteins under the control of a doxycycline-inducible promoter (*Figure 1—figure supplement 2*). We used a biotin-acceptor peptide as a tag, and carried out stringent purifications using streptavidin beads to exclude non-specific co-purification of unrelated SR-proteins, and examined the eluates using immunoblotting. Surprisingly, our results show that the main target of mAb SC35 on these immunoblots is SRSF7 (*Figure 1B*), which, when untagged, runs at approximately ~35 kDa on polyacrylamide gels similar to SRSF2 (the tag combination that we use shifts both proteins by ~15 kDa). To exclude any artefacts that could originate from the use of tagged proteins, we used whole-cell extracts from HEK293 cells and immunoprecipitated targets of mAb SC35 and analyzed the eluates via immunoblotting. Consistent with the results of the IP-MS experiment, and tagged-SRSF1-12 purifications, we observed a very clear enrichment for SRRM2 and SRSF7, but not for SRSF2, SRSF1 or other factors (*Figure 1C*). In order to determine whether the 35 kDa band recognized by mAb SC35 in immunoblots of cellular lysates is composed of multiple proteins, in addition to SRSF7, we created a cell line where we inserted the FKBP12$^{F36V}$ degron tag (*Nabet et al., 2018*) homozygously into the C-terminus of SRSF7 in HEK293 cells. Even without any treatment, it is evident that the 35 kDa band robustly recognized by mAb SC35 in wild-type cells completely disappears in SRSF7$^{FKBP12}$ cells (*Figure 1D*, compare lanes 1 and 2 with 3 and 4), and a new band around ~50 kDa, where the FKBP12$^{F36V}$-tagged SRSF7 runs, emerges (*Figure 1D*, arrowhead). Treatment of these cells with 1 µM of dTAG7 for 6 hr lead to the depletion FKBP12$^{F36V}$-tagged SRSF7, and to the depletion of the newly-emerged ~50 kDa protein recognized by mAb SC35. The identity of a fainter band around 37 kDa, which is insensitive to tagging SRSF7 or dTAG treatment remains unknown. These results strongly suggest that the 35 kDa namesake protein revealed by SC35 mAb on immunoblots is SRSF7 and any contribution to this signal from other proteins is negligible to none.

## SRRM2 is the primary target of mAb SC35 in immunoblots

Even though our results show that SC35 mAb specifically recognizes SRSF7 rather than SRSF2, both proteins have significant nucleoplasmic pools in addition to their localization to NS (*Politz et al., 2006*; *Prasanth et al., 2003*; *Sapra et al., 2009*) which is not easily reconciled with the immunofluorescence stainings obtained with the SC35 mAb that are virtually restricted to NS. Intriguingly, SRRM2, which is by far the most enriched protein in our immunoprecipitations with mAb SC35, is a

relatively large (~300 kDa) protein, that readily co-purifies with spliceosomes (*Bertram et al., 2017*; *Bessonov et al., 2010*), shows liquid-like behaviour in cells (*Rai et al., 2018*) and co-localizes near-perfectly with mAb SC35-stained NS (*Miyagawa et al., 2012*). In addition, SRRM2 and its yeast counterpart Cwc21/Cwf21 is located in the recent cryo-EM structures of the spliceosome, where it joins the spliceosome at the $B^{act}$ stage where it seems to support the activated conformation of PRP8's switch-loop both in humans and yeast (*Jia and Sun, 2018*). Predating the recent cryo-EM structures by almost a decade, the yeast orthologue of SRRM2, Cwf21p, has been shown to directly interact with Prp8p (PRPF8) and Snu114p (EFTUD2) which are also among the most enriched proteins in our mAb SC35 immunoprecipitations (*Grainger et al., 2009*; *Figure 1—figure supplement 1A*). Furthermore, a more recent tandem-affinity purification of the protist *Trypanosoma* orthologue of SRRM2 revealed Prp8, U5-200K (SNRNP200, also known as Brr2), U5-116K (EFTUD2, also known as Snu114) and U5-40K (SNRNP40) as major interaction partners (*Silva et al., 2011*).

Taking into consideration the fact the mAb SC35 was raised against biochemically purified spliceosomes (*Fu and Maniatis, 1990*), together with the aforementioned observations in the scientific literature and our IP-MS results which identified SRRM2 as the top target, we hypothesize that mAb SC35 was most likely raised against SRRM2, and it recognizes SRRM2 in most if not all immunological assays that utilizes mAb SC35 where SRRM2 is not depleted or unintentionally omitted due to technical reasons.

In order to test the veracity of this claim, we designed a series of experiments in human cells. Since, to our knowledge, SC35 mAb has not been shown to recognize SRRM2 on immunoblots, we first created tagged and truncated SRRM2 constructs in living cells. To this end, we generated 11 cell lines that remove between 4 and 2322 amino acids from the SRRM2 protein (full-length: 2752 a. a., numbering from Q9UQ35-1) by inserting a TagGFP2 (referred to as GFP for simplicity) sequence followed by an SV40 polyadenylation signal into 11 positions of the *SRRM2* gene in HAP1 cells using a CRISPR/Cas9-based technique called CRISPaint (*Schmid-Burgk et al., 2016*; *Figure 2A*). The deepest truncation removes 84% of SRRM2, which includes almost all its IDRs, together with two regions enriched for serine and arginine residues, leaving behind 13 RS-dipeptides out of a total of 173 (*Figure 2—figure supplement 1A,B*). The GFP-tagged, in vivo truncated proteins (referred to as **tr**uncations 0 to 10, shortened as tr0 - tr10, *Figure 2B*) are then immunoprecipitated using GFP-trap beads and the eluates were analyzed by immunoblotting. This experiment shows that SC35 mAb indeed recognizes SRRM2 on immunoblots (*Figure 2C*, lane 2). Interestingly, the signal from SC35 mAb remains relatively stable up until SRRM2$^{tr4}$ which removes 868 a.a. from the SRRM2 C-terminus, the signal appears to be reduced in SRRM2$^{tr5}$ which removes 1014 a.a. and becomes completely undetectable from SRRM2$^{tr6}$ onward (*Figure 2C* and *Figure 2—figure supplement 1C*). The same blot was stripped and re-probed with a polyclonal antibody raised against the N-terminus of SRRM2, common to all truncations, which show that SRRM2 is detectable throughout, and thus indicating that the epitope(s) recognized by mAb SC35 reside between amino acids 1,360–1884 of SRRM2.

In order to assess the efficiency and the specificity of the GFP-pull-down, we used a wild-type lysate without any GFP insertion, together with lysates made from SRRM2$^{tr0}$ and SRRM2$^{tr10}$ cells, which served as the negative control, positive control and the deepest truncation (tr10) we generated, respectively. The immunoblot with mAb SC35 once again clearly shows that near-full-length SRRM2$^{tr0}$ is recognized by mAb SC35 to the same extent as the SRRM2 polyclonal antibody, while SRRM2$^{tr10}$ is not detected by mAb SC35 at all but strongly with SRRM2 polyclonal antibody (*Figure 2D*). These blots also show that full-length SRRM2 co-purifies SRRM1 and to a lesser extent RBM25, while both interactions are severely compromised in SRRM2$^{tr10}$. Furthermore, the absence of any signal in SRRM2$^{tr10}$ input lane probed with mAb SC35 (*Figure 2D*, top left lane 3), and the emergence of a shorter ~100 kDa protein in the complete absence of a ~300 kDa signal in the SRRM2 blot (*Figure 2D*, bottom left, lane 3) shows that SRRM2$^{tr10}$ cells have a homozygous insertion of the GFP construct, which was also confirmed by genotyping PCR (*Figure 2—figure supplement 1D*). This result further indicates that the large IDRs of SRRM2 are not essential for cell viability, at least in HAP1 cells.

These results can be puzzling, since we first show that mAb SC35 specifically recognizes a 35 kDa band which we reveal to be SRSF7 (*Figure 1*), but later, in a separate set of experiments, we also show that mAb SC35 specifically recognizes a ~300 kDa band, which we reveal to be SRRM2 (*Figure 2*), while the original study describing mAb SC35 reports a single 35 kDa band recognized by

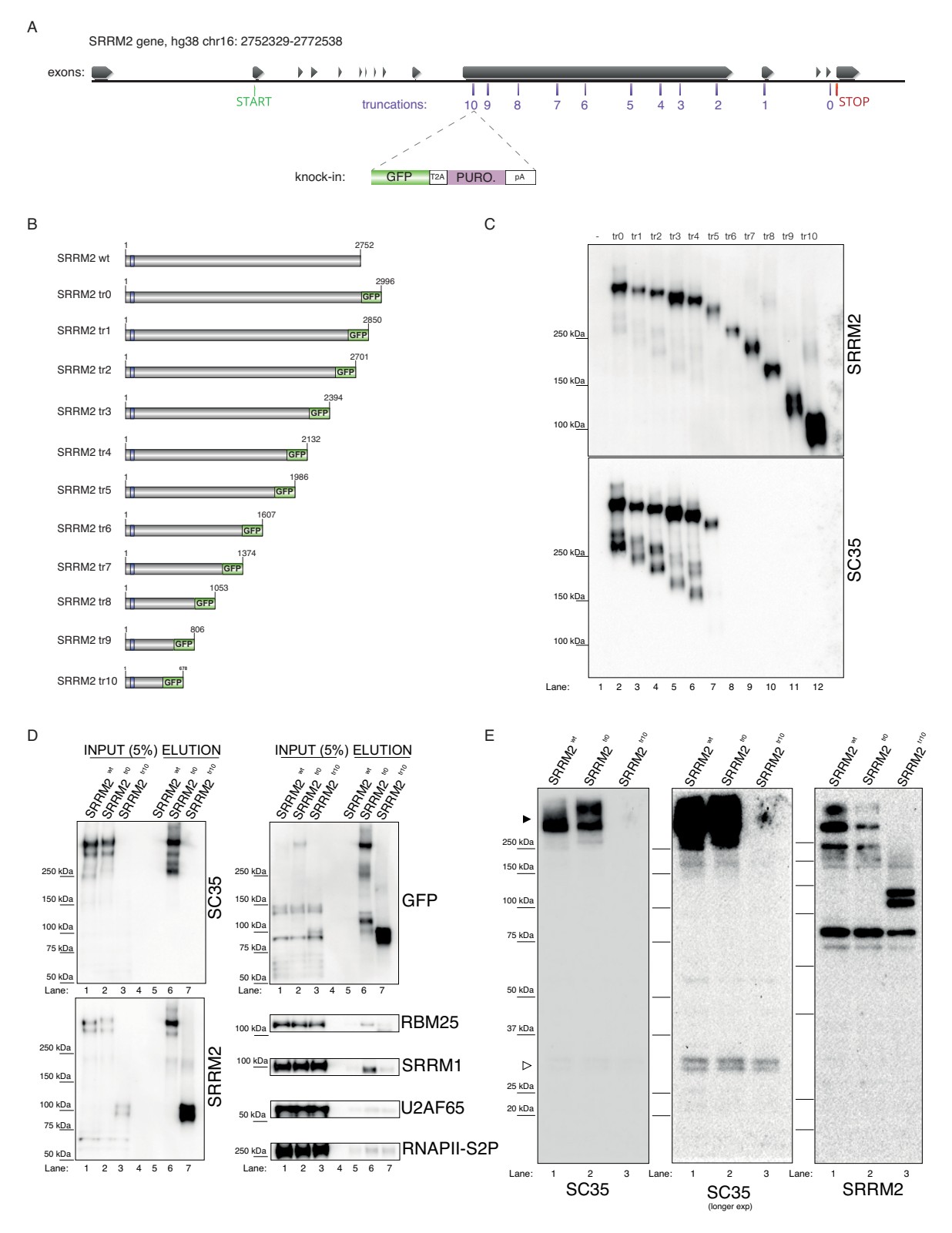

**Figure 2.** Endogenous truncating mutations of SRRM2 prove mAb SC35 as an SRRM2 antibody. (**A**) The strategy for the CRISPaint generated endogenous truncating mutations (0-to-10) accompanied by the TagGFP2 (depicted as GFP for simplicity) fusion are shown. (**B**) The sizes of SRRM2 truncated GFP fusion proteins are displayed. (**C**) Protein purified using a GFP-trap pull-down from lysates of corresponding stable HAP1 cell lines carrying the truncated SRRM2 alleles are run on PAGE. Western blotting of SRRM2 using an antibody generated against the common N-terminus is
*Figure 2 continued on next page*

*Figure 2 continued*
used to show the amount of loaded protein on the gel. SC35 blot shows a significant reduction in signal intensity of SRRM2-tr5 and a complete loss of signal from SRRM2-tr6 to tr10. (D) GFP-trap pull-down performed on lysates from wild-type, tr0 and tr10 HAP1 cells enrich for SRRM2 in tr0 cells, indicating the GFP-tagged allele is specific to SRRM2 and is detected by also SC35 blot (Lanes 1 and 2 inputs compared to Lanes 5 and 6 on the upper and lower left-side blots). SRRM2 also co-purifies two other NS-associated proteins; SRRM1 and RBM25 (Lane 6 on lower right-side blots). SRRM2-tr10 is not detected by SC35 but the pull-down efficiency (Lane 7 on upper left-side blot) and loading is validated by SRRM2 (Lanes 3 and 7 on lower left-side blot) and GFP blots (Lanes 3 and 7 on upper left-side blot). (E) Total cell lysates from wild-type, tr0 and tr10 HAP1 cells are run on 4–12% polyacrylamide gel and blotted with SC35 reveal the high-molecular weight (~300 kDa) as the most intense band and the absence of signal in tr10 cell lines validates that this band represents SRRM2 (filled arrow head). Longer exposure of the blot reveals a weak cross-reactivity with a 35 kDa protein, most likely to be SRSF7, around 35 kDa (empty arrow head).

The online version of this article includes the following figure supplement(s) for figure 2:

**Figure supplement 1.** The strategy for making the truncating mutations of SRRM2.

mAb SC35 on immunoblots (*Fu and Maniatis, 1990*). The solution to this conundrum presented itself in the form of altering the immunoblotting technique. Using whole-cell extracts prepared from wild-type cells, together with SRRM2$^{tr0}$ and SRRM2tr$^{10}$ cells, in a gel system where we can interrogate both small and large proteins simultaneously, we were able to detect both SRRM2 and SRSF7 on the same blot (*Figure 2E*). These blots prove that the ~300 kDa band is indeed SRRM2, since it completely disappears in SRRM2$^{tr10}$ lysates (which is accompanied by the appearance of a ~100 kDa band in SRRM2 blots) while the much fainter 35 kDa band corresponding to SRSF7 (*Figure 1*) remains unaltered.

These experiments provide strong support for our hypothesis that the main target of SC35 mAb is SRRM2, a protein proven to be part of spliceosomes, against which this antibody was raised, and suggests that a cross-reactivity towards SRSF7, likely in combination with immunoblotting techniques not suitable to detect large proteins (*Bass et al., 2017*), obscured this fact for more than two decades.

## SRRM2 is the primary target of mAb SC35 in immunofluorescence stainings

mAb SC35 is typically used as an antibody in immunofluorescence experiments that reveals the location of NS in mammalian cells (*Spector and Lamond, 2011*). In light of the evidence presented here, it can be assumed that mAb SC35 primarily stains SRRM2 in immunofluorescence stainings, as in immunoblotting experiments. In order to test if this is indeed the case, we took advantage of the SRRM2$^{tr10}$ cells. These cells are viable and express a severely truncated SRRM2 that is not recognized by mAb SC35 on immunoblots (*Figure 2*).

SRRM2$^{tr10}$ cells, together with SRRM2$^{tr0}$ cells serving as a control, were stained with antibodies against various nuclear speckle markers, including mAb SC35 (*Figure 3A*). These results show that SC35 signal virtually disappears in SRRM2$^{tr10}$ cells, while other markers of NS, such as SON, SRRM1, and RBM25 appear unaltered, ruling out a general defect in NS (*Figure 3A*). As an additional control, we also mixed SRRM2$^{tr10}$ cells with SRRM2$^{tr0}$ cells together before formaldehyde fixation, and repeated the antibody stainings, in order to be able to image these two cell populations side-by-side. SRRM2$^{tr10}$ and SRRM2$^{tr0}$ cells are easily distinguished from each other since the latter show a typical nuclear speckle staining whereas the former has a more diffuse, lower intensity GFP signal. These images clearly show that mAb SC35, obtained from two separate vendors, no longer stains NS or any other structure in SRRM2$^{tr10}$ cells, whereas other NS markers, including SRSF7, appear unaltered (*Figure 3B,C*, *Figure 3—figure supplement 1*).

Taken together, our results show that mAb SC35, which was raised against a spliceosomal extract, was most likely raised against SRRM2, a ~300 kDa protein that, unlike SRSF2 or SRSF7 is present in spliceosomes of both in yeast and humans. We show that mAb SC35 directly recognizes SRRM2 between amino acids 1,360–1,884, and that the main signal from mAb SC35 corresponds to SRRM2 both in immunoblots and immunofluorescence images. It is important to note that these results were obtained from unsynchronised human cells, which are mostly at the interphase stage of the cell-cycle. mAb SC35 might recognize additional targets in mitotic cells or cells derived from non-mammalian species.

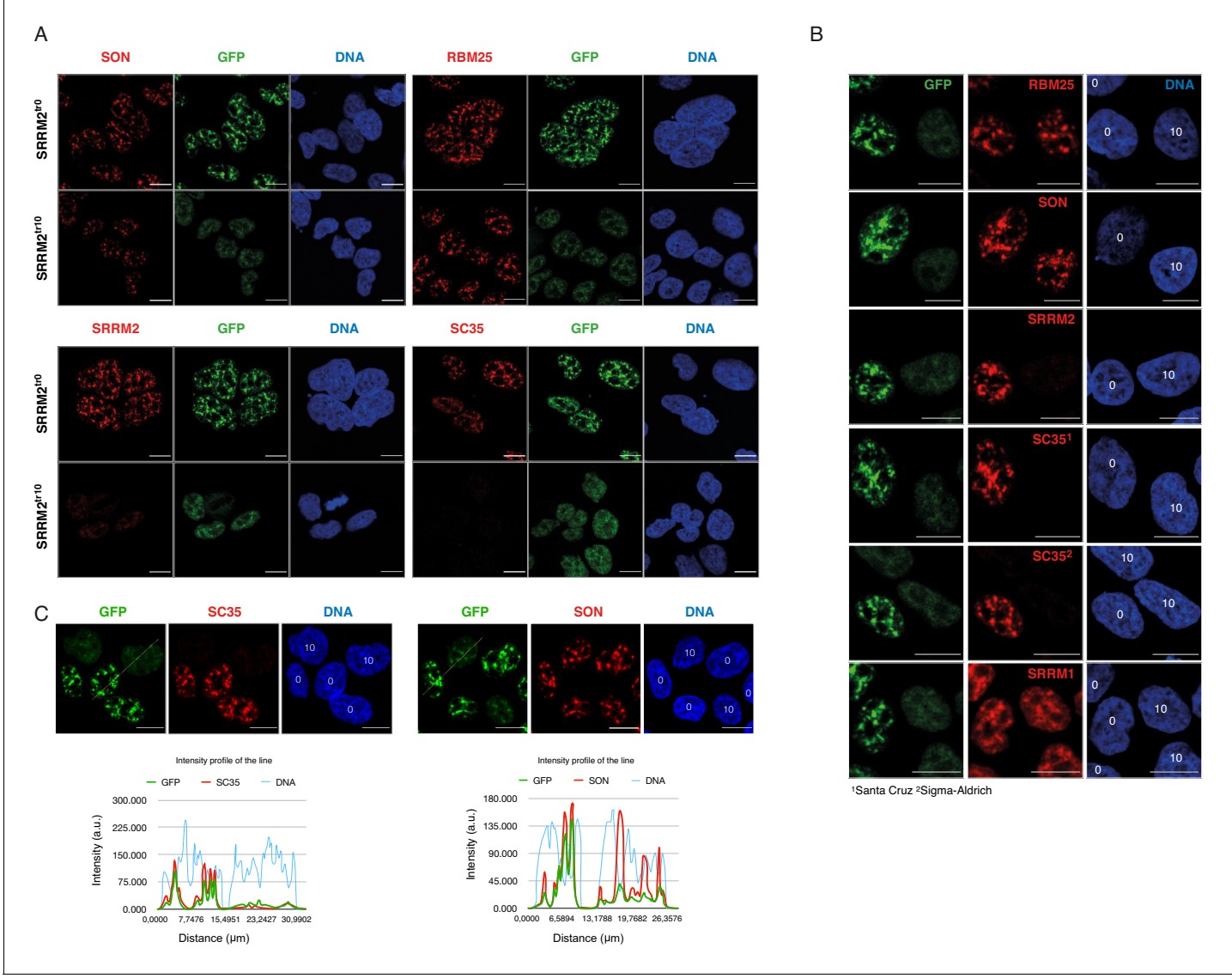

**Figure 3.** SRRM2 truncation#10 leads to loss of SC35 domains but not NS. (**A**) SON and RBM25 antibodies are used as NS markers for IF analysis of both SRRM2$^{tr0}$ and SRRM2$^{tr10}$ HAP1 cells. No significant impact on the formation of NS in SRRM2$^{tr10}$ cells in comparison to SRRM2$^{tr0}$ cells is observed. Lack of signal for SC35 in SRRM2$^{tr10}$ cells validates SC35 as an SRRM2 antibody. (**B**) The SRRM2$^{tr0}$ and SRRM2$^{tr10}$ cells are plated together before the IF protocol is performed and the GFP signal intensity as well as SC35 staining are used to distinguish SRRM2$^{tr10}$ cells from SRRM2$^{tr0}$ HAP1 cells. The DNA stain marking the nuclei are annotated with '0' or '10' on top to indicate the corresponding cell line. (**C**) The SRRM2$^{tr0}$ and SRRM2$^{tr10}$ HAP1 cells are imaged side-by-side and a line is drawn to quantify the signal intensity across two cell lines. The intensity profile of the lines shows dramatically reduced signal for SC35 between SRRM2$^{tr0}$ and SRRM2$^{tr10}$ cells, whereas similar signal intensities for SON and DNA in SRRM2$^{tr0}$ cells is observed. Scale bars = 10 µm.

The online version of this article includes the following figure supplement(s) for figure 3:

**Figure supplement 1.** SRSF7 staining in SRRM2$^{tr0}$ compared to SRRM2$^{tr10}$ cells.

It is also interesting to note that this is not the first time an antibody is serendipitously raised against SRRM2 and was later discovered to recognize SRRM2 only after the fact: In 1994, *Blencowe et al., 1994* reported three murine monoclonal antibodies, B1C8, H1B2, and B4A11 which were raised against nuclear matrix preparations. All three antibodies showed extensive co-localization with NS, although a co-localization between mAb SC35 and B4A11 could not directly be assessed since both mAb SC35 and B4A11 are reported to be IgG mAbs. In a separate work, *Blencowe et al., 1994* showed that B4A11 is an antibody against SRRM2, suggesting that SRRM2 is present both in spliceosomal purifications and nuclear matrix preparations.

## NS formation requires SON and full-length SRRM2

During this work, we noticed the remarkable size difference between human SRRM2 protein (2752 a. a.), and its unicellular counterparts *S. cerevisiae* Cwc21 (133 a.a), *S. pombe* Cwf21 (293 a.a) and *T. Brucei* U5-Cwc21 (143 a.a). Moreover, while all three proteins share a conserved N-terminus, which interact with the spliceosome, the serine and arginine-rich extended C-terminus of human SRRM2 is predicted to be completely disordered (*Figure 2—figure supplement 1B*). Intrigued by this observation, we compiled all metazoan protein sequences of SRRM2, together with SRRM1, RBM25, PNN, SON, PRPF8 and COILIN, and analyzed their size distributions (*Figure 4A*). This analysis confirmed that, unlike SRRM1, RBM25, PNN, PRPF8 or coilin, SRRM2 indeed has a very broad size distribution within metazoa (*Figure 4—figure supplements 1–2*). Strikingly, SON follows this trend with orthologues as small as 610 a.a in the basal metazoan sponge *A. queenslandica*, and as large as 5561 a.a in the frog *X. tropicalis*. Increase in protein size appears to involve IDR extensions, especially for SRRM2, but also for SON (*Figure 4B*, *Figure 4—figure supplements 1–2*), suggesting a role in LLPS-mediated condensate formation, which was shown to be the case for both SRRM2 (*Rai et al., 2018*) and SON (*Kim et al., 2019*) in living cells.

Putting together the observation that places SON and SRRM2 at the center of NS (*Fei et al., 2017*, with the interpretation that SC35 stains SRRM2 in their microscopy work), the presence of SRRM2 at the center of collapsed speckles in SON knock-down experiments, and the peculiar variation in the sizes of SON and SRRM2 during evolution involving gain of IDRs, we hypothesize that SON, together with SRRM2 are essential for NS formation, such that SRRM2 continues to serve as a platform for NS-associated proteins in SON-depleted cells.

In order to test this hypothesis, we used the SRRM2tr0 and SRRM2tr10 cells as a model, which allowed us to simultaneously detect SON, SRRM2 and an additional NS marker in the same cell. We chose SRRM1, which is used as a marker for NS in immunofluorescence experiments

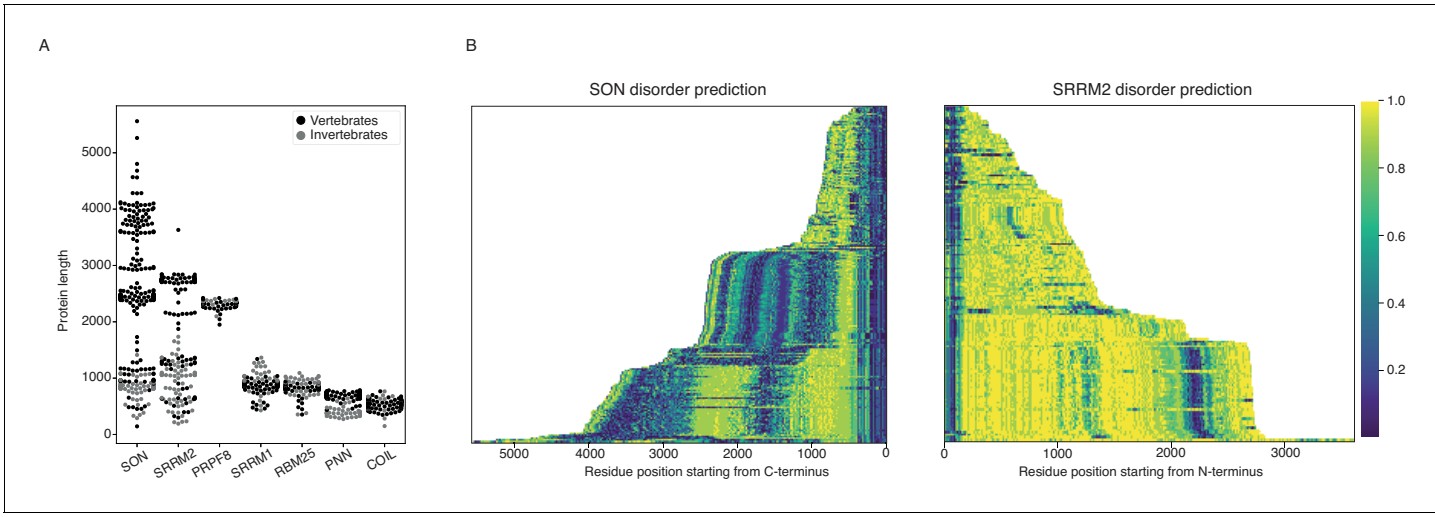

**Figure 4.** SON and SRRM2 are rapidly evolving and largely disordered proteins. (**A**) The size distribution of SON and SRRM2 is highly variable across metazoan species with a mean length of 2227.9 a.a. and SD of 1149.5 for SON and a mean length of 1928.6 a.a. and SD of 919.3 for SRRM2. The lengths of other NS-associated proteins are less variable with a mean length of 895.1 a.a and SD of 104.2 for SRRM1; mean length of 835.4 a.a. and SD of 77.9 for RBM25; mean length of 652.8 a.a. and SD of 118.7 for PNN, mean length of 2332.8 a.a. SD of 40.8 for PRP8. (**B**) The disorder probability of SON and SRRM2 is predicted using the MobiDB-Lite algorithm, which shows an increase of disordered content with the increase of protein length for SRRM2, and to some extent, SON. The SON and SRRM2 graphs plotted side-by-side do not correspond to the same species, for a phylogeny resolved version of this graph see *Figure 4—figure supplement 1* and for the alternative algorithm (IUPred2A) see *Figure 4—figure supplement 2*. The color is scaled from dark blue to yellow indicating a decrease in order as the value approaches 1.0 (yellow).

The online version of this article includes the following source data and figure supplement(s) for figure 4:

**Source data 1.** Contains the numerical values of protein lengths shown in *Figure 4A* and disorder predictions shown in *Figure 4B* and *Figure 4—figure supplements 1* and *2* (.csv) using two alternative algorithms (IUPred2A and MobiDB-Lite).

**Figure supplement 1.** SON and SRRM2 are rapidly evolving and largely disordered proteins.

**Figure supplement 2.** SON and SRRM2 are rapidly evolving and largely disordered proteins.

(*Blencowe et al., 2000*; *Blencowe et al., 1998*; *Blencowe et al., 1994*; *Rai et al., 2018*; *Zhang et al., 2016*) and located at IGCs in electron microscopy experiments (*Wan et al., 1994*; RBM25, which is one of two recommended factors to mark NS by the Human Protein Atlas; '*The Human Protein Atlas version 19.3, 2020*' n.d.; *Thul et al., 2017*) (the other being SRRM2), localizes to NS through its RE/RD-rich mixed-charge domain (*Zhou et al., 2008*) that was recently shown to target proteins to NS (*Greig et al., 2020*) and PNN, which localizes to NS in human cells (*Chiu and Ouyang, 2006*; *Joo et al., 2005*; *Lin et al., 2004*; *Zimowska et al., 2003*).

As reported previously (*Ahn et al., 2011*; *Fei et al., 2017*; *Sharma et al., 2010*), depletion of SON leads to collapsed speckles in SRRM2$^{tr0}$ cells, with SRRM2, SRRM1, PNN, and RBM25 localizing to these spherical NS to different extents (*Figure 5A*, *Figure 5B*, *Figure 5—figure supplement 3*, compare SRRM2$^{tr0}$ cells, control vs SON siRNA treatment). In SRRM2$^{tr10}$ cells on the other hand, where the truncated SRRM2 has a significant nucleoplasmic pool already in control siRNA treated cells, depletion of SON leads to a near-complete diffusion of truncated SRRM2, which is followed by RBM25 (*Figure 5B*), SRRM1 (*Figure 5—figure supplement 3A*) and PNN. Using ilastik and CellProfiler, we quantified the signal detected in NS, and compared it to signal detected in the entire nucleus for each cell in every condition for each protein investigated (*Figure 5—figure supplement 2*). These results show that truncated SRRM2 shows reduced NS localization (*Figure 5C*, right), while RBM25, SRRM1, and PNN are localized at NS to a similar extent in SRRM2$^{tr0}$ and SRRM2tr$^{10}$ cells, although with a broader distribution in SRRM2$^{tr10}$ cells. Depletion of SON in SRRM2$^{tr0}$ leads to a significant reduction in NS localization for all proteins, verifying SON's importance for NS formation. Depletion of SON in SRRM2$^{tr10}$ cells, however, leads to a more dramatic loss of NS localization for all proteins (*Figure 5* and *Figure 5—figure supplement 3*), underscoring the essential role of SRRM2's extended IDR in the formation of NS, especially in SON-depleted cells. Number of Cajal bodies, determined by COILIN staining, remains unaltered in all conditions (*Figure 5—figure supplement 3B*).

Next, to independently verify these observations, we knocked-down SON and SRRM2, individually and simultaneously in HEK293 cells where we endogenously tagged SRRM2 with TagGFP2 at the C-terminus with the same reagents used to create SRRM2$^{tr0}$ HAP1 cells. Similar to the HAP1 model, depletion of SON alone leads to collapsed NS where SRRM2, RBM25, PNN, and SRRM1 localize to spherical NS to some extent but with a significant non-NS pool in the nucleus (*Figure 5—figure supplement 4*). Depletion of SRRM2 alone also leads to delocalization of PNN, SRRM1, and RBM25 from NS, but not to the extent seen with SON depletion. Co-depletion of SON and SRRM2 leads to near-complete delocalization of all proteins investigated, mirroring the results obtained from the HAP1 model (*Figure 5—figure supplement 4A,B,C and D*). These results cannot be explained by reduced protein stabilities, as none of the proteins except for SON and SRRM2 show significant changes in their amounts as judged by immunoblotting (*Figure 5—figure supplement 4E*). Finally, co-depletion of SON together with SRRM1 or RBM25 does not lead to diffusion of spherical NS marked by SRRM2, indicating that SRRM2 has a unique role in NS formation and cannot be substituted by other NS-associated factors (*Figure 5—figure supplement 5*).

## Discussion

Our results have broad implications with respect to the biology in and around NS. One of the primary culprits in the so-called 'reproducibility crisis' in natural sciences is considered to be mischaracterized antibodies (*Baker, 2015*), which led to initiatives to validate them appropriately (*Uhlen et al., 2016*), and large consortia such as ENCODE (*Davis et al., 2018*) publishes specific guidelines for the characterization antibodies that are used to generate data pertinent to the ENCODE project ('*ENCODE, 2020*,' n.d.).

It is therefore reasonable to suspect that the dissonance between observations made with mAb SC35 and subsequent observations made with SRSF2 reagents could have led to misinterpretation of vast amounts of primary data. It is beyond the scope of this work to review each and every study that has used this antibody in the last 30 years, however, since both *SRSF2* and *SRRM2* are clinically important genes (*Anczuków and Krainer, 2016*; *Inoue et al., 2016*; *Meggendorfer et al., 2012*; *Shehadeh et al., 2010*; *Tanaka et al., 2018*; *Tomsic et al., 2015*; *Yoshimi et al., 2019*), and SRRM2 gene is highly to intolerant to loss-of-function mutations in human populations (expected/observed ratio for *SRRM2* is 6%, median expected/observed ratio for all gene variants is 48%)

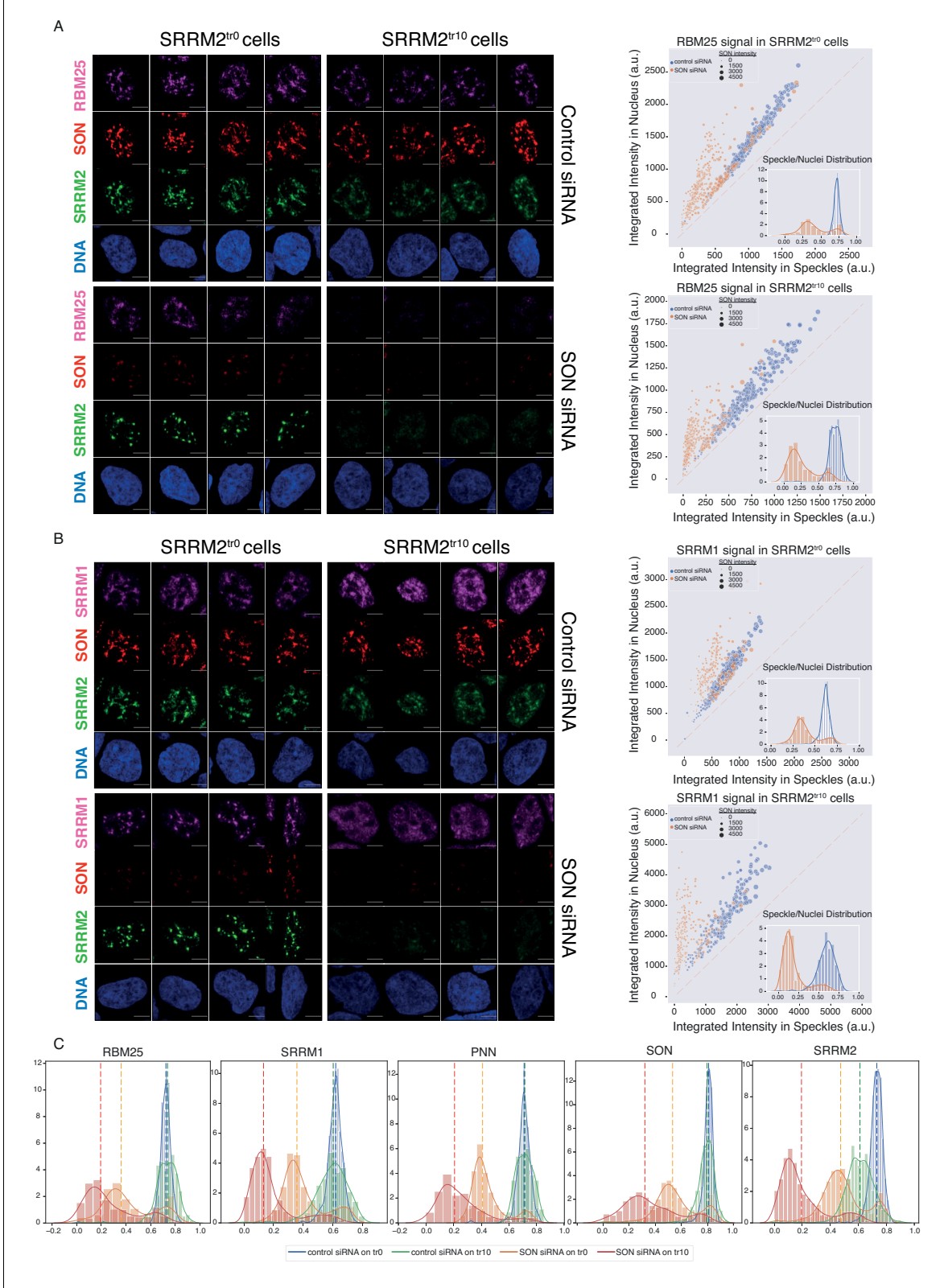

**Figure 5.** SON and SRRM2 form NS in human cells. (**A**) RBM25 IF signal is shown for four individual cells in each siRNA treatment (control or SON siRNA) in SRRM2$^{tr0}$ and SRRM2$^{tr10}$ HAP1 cells. The NS localization of RBM25 is severely reduced upon SON knock-down in SRRM2$^{tr0}$ cells, and completely lost upon SON knock-down in SRRM2$^{tr10}$ cells. The quantification of the RBM25 signal within the nucleus is plotted against the RBM25 signal within NS (right panel) using ilastik to train detection of NS and CellProfiler for quantification on 10 imaged fields with a 63X objective (in

*Figure 5 continued on next page*

*Figure 5 continued*

SRRM2$^{tr0}$ cells control n = 329, SON-KD n = 422; in SRRM2$^{tr10}$ cells control n = 329, SON-KD n = 402). Each circle represents a cell and the size of the circles is proportionate to the signal intensity of SON. Inset shows the distribution of the ratio of signal detected in NS over signal detected in the nucleus of each cell. (B) SRRM1 IF signal is shown for four individual cells in each siRNA treatment (control or SON siRNA) in SRRM2$^{tr0}$ and SRRM2$^{tr10}$ HAP1 cells. The NS localization of SRRM1 is reduced in SON knock-down in SRRM2$^{tr10}$ cells and lost upon SON knock-down in SRRM2$^{tr10}$ cells. The quantification of the SRRM1 signal within the nucleus is plotted against the SRRM1 signal within NS (right panel) using ilastik to train detection of NS and CellProfiler for quantification on 10 imaged fields with 63X objective (in SRRM2$^{tr0}$ cells control n = 494, SON-KD n = 229; in SRRM2$^{tr10}$ cells control n = 225, SON-KD n = 247). Inset shows the distribution of the ratio of signal detected in NS over signal detected in the nucleus of each cell. (C) Distribution plots showing the ratio of signal detected in NS over signal detected in the nucleus of each cell, in each condition. The dashed line indicates the median ratio in each condition. See *Figure 5—figure supplement 3A* for a full version of this analysis for PNN. Scale bars = 5 μm. The online version of this article includes the following source data and figure supplement(s) for figure 5:

**Source data 1.** Contains two folders, 'NS' and 'Cajal'.
**Figure supplement 1.** Training a machine learning method for detection of NS.
**Figure supplement 2.** Examples of the trained module for detection of NS on different antibody stainings indicate the model predicts NS robustly for each stained protein.
**Figure supplement 3.** Depletion of SON in SRRM2$^{tr10}$ cells leads to loss of NS but not of Cajal bodies.
**Figure supplement 4.** Co-depletion of SON and SRRM2 in SRRM2$^{tr0}$+GFP HEK293 cells leads to loss of NS.
**Figure supplement 4—source data 1.** Contains ilastik models (.ilp) used to train for nuclear speckles, and Cell Profiler pipelines (.cpproj) used to process the probability maps generated by ilastik.
**Figure supplement 5.** Co-depletion of SON with RBM25 or SRRM1 in SRRM2$^{tr0}$ HAP1 cells does not lead to loss of spherical NS.

(*Karczewski et al., 2020*), the opportunity cost incurred when results generated with mAb SC35 have been misinterpreted must be carefully considered.

For example, GSK-3, a protein kinase implicated in Alzheimer's disease (*Bhat et al., 2018*), has been shown to phosphorylate SRSF2 (*Hernández et al., 2004*). This study reported that inhibition of GSK-3 leads to spherical NS in cortical neurons, which was interpreted as SRSF2 accumulating at NS upon GSK-3 inhibition. Based on this assumption, a cerebral cortex lysate was immunoprecipitated with mAb SC35, and a 35 kDa band was shown to be phosphorylated by recombinant GSK-3, whereas neither SRSF7 nor SRRM2 were pursued as potential candidates. Interestingly, a more recent study reported that SRRM2 is a target of GSK-3 (*Shinde et al., 2017*). In another study, mAb SC35 has been used as a reporter for phosphorylated SRSF2 in immunohistochemical analysis of tissue samples obtained from patients with Non Small Cell Lung Carcinoma (*Gout et al., 2012*), and was reported to be upregulated in cancer tissues compared to normal tissues. SRRM2's potential role in these syndromes therefore remains unexplored.

In a more non-clinical setting, recent microscopy work suggests that RNA polymerase II, through its intrinsically disordered CTD, switches from transcriptional condensates to NS during progression from initiation to productive elongation (*Guo et al., 2019*), which is in line with recent work that links NS to augmented gene expression (*Chen and Belmont, 2019*). In this particular work, initial observations made with the SC35 mAb have been followed by fluorescently tagged SRSF2 protein for live-cell imaging analysis and condensates formed by SRSF2 are used as a surrogate for splicing condensates in vitro. Furthermore, ChIP-seq profiles generated with mAb SC35 were interpreted as SRSF2 occupancy in vivo, which was shown to be enriched at the 3'-ends of genes, together with serine-2 phosphorylated RNAPII. SRRM2's potential role in these processes remains unknown. In another study, loss of mAb SC35 signal in cells treated with siRNAs against SRRM2 has been interpreted to be a sign of NS disassembly (*Miyagawa et al., 2012*), and based on this knowledge, a more recent study interrogated the 3D conformation of the mouse genome in SRRM2-depleted cells using Hi-C (*Hu et al., 2019*), results of which are interpreted to be a consequence of NS disruption. A recent super resolution microscopy study made use of the mAb SC35 to show that SRSF2 is at the core of NS together with SON, and remains at the core after depletion of SON (*Fei et al., 2017*). In this study, a minimalist computational model was used to model the distribution of five NS-associated factors (SON, SRSF2 on account of SC35 mAb stainings, MALAT1, U1, and U2B'') which could have benefited from the knowledge that SC35 mAb stains SRRM2 rather than SRSF2, considering that pairwise interactions between SRRM2 and other components would be drastically different to SRSF2's potential interactions due to the extensive IDRs of SRRM2 compared to SRSF2. This study

also highlights the fact that a crucial aspect of nuclear speckle biology, namely whether NS are nucleated by specific factors or not, remained an open question.

In contrast, many membraneless organelles have been shown to depend on a small number of factors which act as scaffolds or nucleation points for their formation. Paraspeckles for instance require lncRNA NEAT1, without which paraspeckles do not form (*Chen and Carmichael, 2009*), Cajal bodies are disrupted or disappear in the absence of COILIN, SMN, FAM118B or WRAP53 (*Li et al., 2014*; *Mahmoudi et al., 2010*), and PML bodies are nucleated by PML (*Lallemand-Breitenbach and de Thé, 2010*). In the same vein, SRRM2 has been suggested to be essential for the formation of NS (*Miyagawa et al., 2012*), however, this idea was based on the disappearance of mAb SC35 signal in cells transfected with siRNAs against SRRM2, which is the expected result taking the evidence presented here into account, but does not prove that SRRM2 is essential or important for NS formation. Other candidates that were put forward as essential or important for the formation of NS include lncRNA MALAT1 (*Nakagawa et al., 2012*; *Tripathi et al., 2010*), SRSF1 (*Tripathi et al., 2012*), PNN, and SON (*Ahn et al., 2011*; *Fei et al., 2017*; *Sharma et al., 2010*), all of which, with the exception of PNN, lead to the formation of 'collapsed' speckles rather than a bulk release of NS-associated factors and their diffusion into the nucleoplasm, which would indicate a true loss of NS. Depletion of PNN was shown to either lead to 'collapsed' speckles (*Joo et al., 2005*) or to loss of NS altogether, but under conditions that also lead to degradation of all SR-proteins tested in that particular study (*Chiu and Ouyang, 2006*). To our knowledge, NS could only be successfully dissolved by overexpression of CLK1/STY kinase, which phosphorylates SR-proteins (*Sacco-Bubulya and Spector, 2002*), DYRK3, another protein kinase that can dissolve multiple membraneless bodies (*Rai et al., 2018*), overexpression of PPIG, a peptidyl-proline isomerase (*Lin et al., 2004*) or more recently by overexpression of TNPO3, which is an import factor that binds to phosphorylated SR-residues (*Hochberg-Laufer et al., 2019*). Such observations and lack of evidence to the contrary, led to the idea that NS formation happens through stochastic self-assembly of NS-associated factors, without the need for an organizing core (*Dundr and Misteli, 2010*; *Spector and Lamond, 2011*; *Tripathi et al., 2012*). In this work, we show that the spherical bodies left over after depletion of SON, which are strongly stained with mAb SC35, and also by other NS markers such as SRRM1, PNN, RBM25 can be dissolved by either co-depleting SRRM2 together with SON, or depleting SON in a cell-line where we deleted the intrinsically disordered C-terminus of SRRM2. Co-depletion of SON together with either SRRM1 or RBM25, two relatively large proteins that also possess prominent intrinsically disordered regions and localize to NS, does not lead to the dissolution of leftover NS, which remain as spherical bodies strongly stained with SRRM2. Since SON is essential for mitosis (*Sharma et al., 2010*), conditions reported here therefore open an approximately 24 hr window to study the transcriptional and post-transcriptional effects of lacking NS in human cells.

## Conclusions

Taken together, our results show that a widely-used monoclonal antibody to mark NS, SC35 mAb, was most likely raised against SRRM2 and not against SRSF2 as it was initially reported. We speculate that this mischaracterization hindered the identification of the core of NS, without which NS do not form, which we show to consist most likely of SON and SRRM2. We found that these two factors, unlike other splicing related proteins analyzed, have gone through a remarkable length extension through evolution of metazoa over the last ~0.6–1.2 billion years, mostly within their IDRs which are typically involved in LLPS and formation of biomolecular condensates. The exact mechanism of NS formation by SON together with SRRM2, and the evolutionary forces that led to the dramatic changes in their lengths remain to be discovered.

## Materials and methods

**Key resources table**

| Reagent type (species) or resource | Designation | Source or reference | Identifiers | Additional information |
|---|---|---|---|---|
| Gene (*Homo sapiens*) | SRRM2 | NCBI | Gene ID: 23524 | |
| Gene (*Homo sapiens*) | SRSF7 | NCBI | Gene ID: 6432 | |

*Continued on next page*

*Continued*

| Reagent type (species) or resource | Designation | Source or reference | Identifiers | Additional information |
|---|---|---|---|---|
| Gene (*Homo sapiens*) | SON | NCBI | Gene ID: 6651 | |
| Cell line (*Homo sapiens*) | HAP1 | Horizon | Cat. #: C631 | |
| Cell line (*Homo sapiens*) | Flp-In T-REx HEK293 | Thermo Fisher Scientific | Cat. #: R78007, RRID:CVCL_U427 | |
| Antibody | SC-35 (Mouse monoclonal) | Sigma-Aldrich (Merck) | Cat. #: S4045, RRID:AB_47751 | IF(1:200) WB(1:1000) |
| Antibody | SC-35 (Mouse monoclonal) | Santa Cruz Biotechnology | Cat. #: sc-53518, RRID:AB_671053 | IF(1:100) |
| Antibody | SRRM2 (Rabbit polyclonal) | Thermo Fisher Scientific | Cat. #: PA5-66827, RRID:AB_2665182 | IF(1:100) WB(1:1000) |
| Antibody | SON (Mouse monoclonal) | Santa Cruz Biotechnology | Cat. #: sc-398508 RRID:AB_2868584 | IF(1:100) |
| Antibody | SON (Rabbit polyclonal) | Sigma-Aldrich (Merck) | Cat. #: HPA023535, RRID:AB_1857362 | IF(1:200) WB(1:1000) |
| Antibody | RBM25 (Rabbit polyclonal) | Sigma-Aldrich (Merck) | Cat. #: HPA070713, RRID:AB_2686302 | IF(1:200) WB(1:1000) |
| Antibody | SRRM1 (Rabbit polyclonal) | abcam | Cat. #: ab221061, RRID:AB_2683778 | IF(1:600) WB(1:2000) |
| Antibody | PNN (Rabbit polyclonal) | abcam | Cat. #: ab244250, RRID:AB_2868585 | IF(1:200) WB(1:1000) |
| Antibody | coilin (Rabbit monoclonal) | Cell Signaling | Cat. #: 14168, RRID:AB_2798410 | IF(1:800) WB(1:2000) |
| Antibody | GFP (Rabbit polyclonal) | Chromotek | Cat. #: PAGB1, RRID:AB_2749857 | WB(1:1000) |
| Antibody | SRSF7 (Rabbit polyclonal) | MBL | Cat. #: RN079PW, RRID:AB_11161213 | IF(1:200) WB(1:1000) |
| Antibody | SRSF1 (Mouse monoclonal) | Santa Cruz Biotechnology | Cat. #: sc-33652, RRID:AB_628248 | WB(1:1000) |
| Antibody | SRSF2 (Rabbit monoclonal) | Abcam | Cat. #: ab28428, RRID:AB_777854 | WB(1:1000) |
| Antibody | U1-70K (Mouse monoclonal) | Santa Cruz Biotechnology | Cat. #: sc-390899, RRID:AB_2801569 | WB(1:1000) |
| Antibody | U2AF65 (Mouse monoclonal) | Santa Cruz Biotechnology | Cat. #: sc-53942, RRID:AB_831787 | WB(1:1000) |
| Antibody | DHX9 (Rabbit monoclonal) | abcam | Cat. #: ab183731, RRID:AB_2868586 | WB(1:1000) |
| Antibody | ADAR (Mouse monoclonal) | Santa Cruz Biotechnology | Cat. #: sc-73408, RRID:AB_2222767 | WB(1:1000) |
| Antibody | Tubulin (Mouse monoclonal) | Santa Cruz Biotechnology | Cat. #: sc-32293, RRID:AB_628412 | WB(1:2000) |
| Antibody | Myc (Rabbit monoclonal) | Cell Signaling | Cat. #: 2276, RRID:AB_331783 | WB(1:1000) |
| Antibody | FLAG (Mouse monoclonal) | Sigma-Aldrich (Merck) | Cat. #: F3165, RRID:AB_259529 | IF(1:200) WB(1:2000) |
| Antibody | RNAPII-S2P (Rabbit monoclonal) | Cell Signaling | Cat. #: 13499, RRID:AB_2798238 | WB(1:1000) |
| Recombinant DNA reagent | CRISPaint Gene Tagging Kit | Addgene | Cat. #: 1000000086, RRID:Addgene_1000000086 | |
| Sequence-based reagent | RBM25 siRNA | Thermo Fisher Scientific | Cat. #: s33912 | 10 nM |
| Sequence-based reagent | SRRM1 siRNA | Thermo Fisher Scientific | Cat. #: s20020 | 10 nM |
| Sequence-based reagent | SON siRNA | Thermo Fisher Scientific | Cat. #: s13278 | 10 nM |
| Sequence-based reagent | SRRM2 siRNA | Thermo Fisher Scientific | Cat. #: s24004 | 10 nM |
| Commercial assay or kit | Pierce MS- Compatible Magnetic IP Kit (Protein A/G) | Thermo Fisher Scientific | Cat. #: 90409 | |
| Commercial assay or kit | Lipofectamine RNAiMAX Reagent | Thermo Fisher Scientific | Cat. #: 13778075 | |
| Chemical compound, drug | dTAG-7 | Tocris | Cat. #: 6912 | 1 μM |
| Software, algorithm | ilastik | https://www.ilastik.org/ | RRID:SCR_015246 | |
| Software, algorithm | CellProfiler | https://cellprofiler.org/ | RRID:SCR_007358 | |

*Continued on next page*

*Continued*

| Reagent type (species) or resource | Designation | Source or reference | Identifiers | Additional information |
|---|---|---|---|---|
| Software, algorithm | Jupyter Lab | https://github.com/jupyterlab/jupyterlab; *Kluyver, 2016* | RRID:SCR_018315 | |

## Cell culture and generation of stable cell lines

Flp-In T-REx HEK293 (Thermo Fisher Scientific Catalog Number: R78007) cells were cultured according to manufacturer's recommendations. The cells were cultured in DMEM with Glutamax supplemented with Na-Pyruvate and High Glucose (Thermo Fisher Scientific Catalog Number: 31966–021) in the presence of 10% FBS (Thermo Fisher Scientific Catalog Number: 10270106) and Penicillin/Streptomycin (Thermo Fisher Scientific Catalog Number: 15140–122). Before the introduction of the transgenes cells were cultured with a final concentration of 100 µg/mL zeocin (Thermo Fisher Scientific Catalog Number: R250-01) and 15 µg/mL blasticidin (Thermo Fisher Scientific Catalog Number: A1113903). To generate the stable cell lines pOG44 (Thermo Fisher Scientific Catalog Number: V600520) was co-transfected with pcDNA5/FRT/TO (Thermo Fisher Scientific Catalog Number: V652020) containing the gene of interest (GOI are SRSF1 to 12 in this case) in a 9:1 ratio. Cells were transfected with Lipofectamine 2000 (Thermo Fisher Scientific Catalog Number: 11668019) on a 6-well plate format with total 1 µg DNA (i.e. 900 ng of pOG44 and 100 ng of pcDNA5/FRT/TO+GOI) according to the transfection protocol provided by the manufacturer. 24 hr after the transfection cells were split on 3 wells of a 6-well plate at 1:6, 2:6 and 3:6 dilution ratios to allow efficient selection of Hygromycin B (Thermo Fisher Scientific Catalog Number: 10687010). The Hygromycin selection was started at the 48 hr after transfection time point with a final concentration of 150 µg/mL and refreshed every 3–4 days until the control non-transfected cells on a separate plate were completely dead (takes approximately 3 weeks from the start of transfection until the cells are expanded and frozen). Induction of the transgene was done over-night with a final concentration of 0.1 µg/mL doxycycline. The cells were validated by performing immunofluorescence by FLAG antibody and western blotting of nuclear and cytoplasmic fractions.

Human HAP1 parental control cell line was purchased from Horizon (Catalog Number: C631) and cultured according to the instructions provided by the manufacturer. The cells were cultured in IMDM (Thermo Fisher Scientific Catalog Number: 12440–053) in the presence of 10% FBS (Thermo Fisher Scientific Catalog Number: 10270106) and Penicillin/Streptomycin (Thermo Fisher Scientific Catalog Number: 15140–122). See *Supplementary file 1* for the list of sgRNAs used in generation of cell lines with CRISPaint. Cells were co-transfected with three plasmids according to the CRISPaint protocol. Cas9 and sgRNA are provided by same plasmid in 0.5 µg final amount, Frame selector plasmid (depending on the cut site selector 0, +1 or +2 had to be chosen) is also in 0.5 µg final amount, the TagGFP2_CRISPaint plasmid was provided at a 1 µg final amount. Therefore the total 2 µg DNA was transfected into cells on 6-well plate format using Lipofectamine 2000. 24 hr after the transfection the cells were expanded on 10 cm culture plates to allow efficient Puromycin (Thermo Fisher Scientific Catalog Number: A1113803) selection. The Puromycin selection is provided in the tag construct and is driven by the expression from the gene locus (in this case the human *SRRM2* gene locus). Puromycin selection was started at 48 hr after transfection at 1 µg/mL final concentration and was refreshed every 2 days and in total was kept for 6 days. After the colonies grew to a visible size the colonies were picked by the aid of fluorescence microscope EVOS M5000. PCR screening of the colonies was performed using genotyping oligos listed in *Supplementary file 1* using Quick Extract DNA Extraction Solution (Lucigen Catalog Number: QE09050) according to manufacturer's protocol in a PCR machine and DreamTaq Green Polymerase (Thermo Fisher Scientific Catalog Number: K1081) using 58˚C annealing temperature and 1 min extension time.

SRSF7-FKBP12$^{F36V}$ knock-in cells were generated in HEK293T cells (ordered from ATCC, CRL3216 and cultured according to the protocol provided) by co-transfecting the sgRNA, Frame selector and mini-circle constructs prepared according to the CRISPaint protocol using Lipofectamine 2000 on a 6-well plate format. This time we used two separate tag donor plasmids to increase the chances of obtaining homozygous clones. The constructs were identical except for the selection antibiotic. Cells are expanded on 10 cm culture plates 24 hr after transfection. At 48 hr after

transfection the double selection was initiated. One allele was selected by Puromycin at 1 μg/mL final concentration, whereas the other allele was selected by blasticidin at 15 μg/mL final concentration for 6 days in total. After the removal of selection cells were kept on the same plate until there were big enough colonies. Colonies were picked under a sterile workbench and screened for homozygosity using western blotting with SRSF7 antibody. The degradation of tagged-SRSF7 was induced by adding dTAG7 reagent at a final 1 μM concentration and keeping for 6 hr.

SRRM2$^{tr0}$-GFP Flp-In TREx HEK293 cells were generated using the same strategy as described above for HAP1 cells. Upon Puromycin selection cells were used as a pool (without sorting or colony picking) in immunofluorescence experiments.

Cell lines are regularly checked for the absence of Mycoplasma using a PCR based detection kit (Jena Biosciences PP-401).

## siRNA transfections

Prior to the seeding of cells, the round glass 12 mm coverslips are coated with poly-L-Lysine hydrobromide (Sigma P9155) for HEK293 cells. The coating is not necessary for the imaging of HAP1 cells. For 1 day of knock-down 40,000 cells are plated on coverslips placed into the wells of 24-well plates on the day before the siRNA transfections. Pre-designed silencer select siRNA (Ambion) are ordered for SRRM2 (ID: s24004), SON (ID: s13278), SRRM1 (ID: s20020) and RBM25 (ID: s33912). Negative control #1 of the silencer select was used for control experiments. 5 nM (for double transfections) or 10 nM (for single transfections) of each siRNA is forward transfected using Lipofectamine RNAiMAX Reagent (Thermo Fisher Scientific Catalog Number: 13778075) according to manufacturer's instructions. The cells were fixed for imaging 24 hr after transfection.

## Immunofluorescence and imaging

### Sample preparation

Cells on coverslips were washed with PBS and crosslinked with 4% paraformaldehyde in PBS (Santa Cruz Biotechnology, sc-281692) for 10 min at room temperature, and washed three times with PBS afterwards. Permeabilization was carried out with 0.5% Triton-X in PBS, 10 min at RT. Cells were washed twice with 0.1% Triton-X in PBS and blocked with 3% BSA (constituted from powder BSA, Roche Fraction V, sold by Sigma Catalog Number: 10735078001) in PBS for 30 min at RT. Primary antibodies were diluted in 3% BSA in PBS, and cells were incubated with diluted primary antibodies for ~16 hr at 4°C in a humidified chamber. Cells were then washed three times with 0.1% Triton-X in PBS and incubated with fluorescently labelled secondary antibodies, diluted 1:500 in 3% BSA for 1 hr at RT, and washed three times with 0.1% Triton-X in PBS. To counterstain DNA, cells were incubated with Hoechst 33258 (1μg/mL, final) for 5 min at RT, and washed once with PBS. Coverslips are briefly rinsed with distilled water and mounted on glass slides using Fluoromount-G (SouthernBiotech, 0100–01) and after a few hours, sealed with CoverGrip (Biotium, #23005) and left in a dark chamber overnight before imaging.

### Antibodies

COIL (Cell Signaling Technology, D2L3J, #14168), FLAG-M2 (Sigma, F3165), PNN (Abcam, ab244250), RBM25 (Sigma, HPA070713-100UL), SC-35 (Santa Cruz Biotechnology, sc-53518), SC-35 (Sigma, S4045), SON (polyclonal rabbit, Sigma, HPA023535), SON (monoclonal mouse, Santa Cruz sc-398508), SRRM1 (Abcam, ab221061), SRRM2 (Thermo Fisher Scientific, PA5-66827).

### Imaging

Images were acquired with a Zeiss LSM880 microscope equipped with an AiryScan detector, using the AiryScan Fast mode with the Plan-Apochromat 63x/1.40 Oil DIC M27 objective. The dimensions of each image were 134 x 134 x 4 μm (Width x Height x Depth), 20 z-stacks were acquired for each image with a step size of 200 nm. Maximum Intensity Projections were created using Zen software (Zeiss) and used for further analysis.

### Analysis

Nuclear speckle identification, segmentation and intensity calculations were carried out using ilastik and CellProfiler. Briefly, eight images were used to train a model that demarcates NS using ilastik

(v.1.3.3post2). CellProfiler was then used to segment nuclei and NS using the probability maps created for each image by ilastik. The data was then analyzed in a Jupyter Lab environment using pandas, SciPy, NumPy and plotted with matplotlib and seaborn. Raw imaging data, models used to train the images, CellProfiler pipelines, and Jupyter notebooks are available.

## Mass spectrometry

### Sample preparation

Pierce MS-Compatible Magnetic IP Kit (Protein A/G) (Thermo Fisher Scientific, Catalog Number: 90409) was used to prepare samples for mass-spectrometry according manufacturer's instructions, where approximately 15 million HAP1 cells (~80% confluent 15 cm dishes) were used per IP. Briefly, HAP1 cells were trypsinized, washed with ice-cold PBS and re-suspended with 500 µL of 'IP-MS Cell Lysis Buffer' which was supplemented with 1x cOmplete Protease Inhibitor Cocktail (Roche, 11697498001) and 1x PhosSTOP (Roche, 4906845001). Cells were then homogenized using a Bioruptor Plus sonifier (30 s ON, 30 s OFF, five cycles on HI). Remaining cellular debris was removed by centrifugation at 21.130 *rcf* for 10 min at 4°C, supernatants were transferred to fresh tubes. 2.5 µL of SC35 mAb (Sigma-Aldrich, S4045) and 25 µL of control IgG1 (Santa Cruz, sc-3877) was used for the SC35 and control IP samples (3 each), respectively and immune-complexes are allowed to form overnight (~16 hr) in the cold-room (~6°C) with end-to-end rotation. Next morning, lysates were incubated with 25 µL of Protein A/G beads for 1 hr in the cold-room, the beads were then washed with 500 µL of ice-cold 50 mM Tris.Cl pH 7.4, 100 mM NaCl, 0.1% Tween-20. The beads were resuspended with the same buffer supplemented with RNaseI (Ambion, AM2295, final concentration 0.02 U/µL) and incubated at 37°C for 5 min. The beads were then washed three times with 'Wash A (+10 mM MgCl2)' and twice with 'Wash B' buffer.

### On beads digest and mass-spectrometry analysis

The buffer for the three SC35 samples and controls was exchanged with 100 µL of 50 mM $NH_4HCO_3$. This was followed by a tryptic digest including reduction and alkylation of the cysteines. Therefore, the reduction was performed by adding tris(2-carboxyethyl)phosphine with a final concentration of 5.5 mM at 37°C on a rocking platform (500 rpm) for 30 min. For alkylation, chloroacetamide was added with a final concentration of 24 mM at room temperature on a rocking platform (500 rpm) for 30 min. Then, proteins were digested with 200 ng trypsin (Roche, Basel, Switzerland) shaking at 600 rpm at 37°C for 17 hr. Samples were acidified by adding 2.2 µL 100% formic acid, centrifuged shortly, and placed on the magnetic rack. The supernatants, containing the digested peptides, were transferred to a new low protein binding tube. Peptide desalting was performed according to the manufacturer's instructions (Pierce C18 Tips, Thermo Scientific, Waltham, MA). Eluates were lyophilized and reconstituted in 11 µL of 5% acetonitrile and 2% formic acid in water, briefly vortexed, and sonicated in a water bath for 30 s prior injection to nano-LC-MS/MS.

### LC-MS/MS instrument settings for shotgun proteome profiling and data analysis

LC-MS/MS was carried out by nanoflow reverse-phase liquid chromatography (Dionex Ultimate 3000, Thermo Scientific) coupled online to a Q-Exactive HF Orbitrap mass spectrometer (Thermo Scientific), as reported previously (*Gielisch and Meierhofer, 2015*). Briefly, the LC separation was performed using a PicoFrit analytical column (75 µm ID ×50 cm long, 15 µm Tip ID; New Objectives, Woburn, MA) in-house packed with 3 µm C18 resin (Reprosil-AQ Pur, Dr. Maisch, Ammerbuch, Germany). Peptides were eluted using a gradient from 3.8% to 38% solvent B in solvent A over 120 min at 266 nL per minute flow rate. Solvent A was 0.1% formic acid and solvent B was 79.9% acetonitrile, 20% $H_2O$, 0.1% formic acid. Nanoelectrospray was generated by applying 3.5 kV. A cycle of one full Fourier transformation scan mass spectrum (300–1750 m/z, resolution of 60,000 at m/z 200, automatic gain control (AGC) target $1 \times 10^6$) was followed by 12 data-dependent MS/MS scans (resolution of 30,000, AGC target $5 \times 10^5$) with a normalized collision energy of 25 eV. To avoid repeated sequencing of the same peptides, a dynamic exclusion window of 30 s was used.

Raw MS data were processed with MaxQuant software (v1.6.0.1) and searched against the human proteome database UniProtKB with 21,074 entries, released in December 2018. Parameters of Max-Quant database searching were a false discovery rate (FDR) of 0.01 for proteins and peptides, a

minimum peptide length of seven amino acids, a first search mass tolerance for peptides of 20 ppm and a main search tolerance of 4.5 ppm. A maximum of two missed cleavages was allowed for the tryptic digest. Cysteine carbamidomethylation was set as a fixed modification, while N-terminal acetylation and methionine oxidation were set as variable modifications. The MaxQuant processed output files can be found in *Supplementary file 2*, showing peptide and protein identification, accession numbers, % sequence coverage of the protein, and q-values.

The mass-spectrometry proteomics data have been deposited to the ProteomeXchange Consortium via the PRIDE partner repository with the dataset identifier PXD021814.

## Pull-downs and immunoblotting

Streptavidin-pulldowns (*Figure 1B*) were carried out using stable-cell-line expression SRSF1-12 proteins. Briefly, for each cell line, ~1 million cells (one well of a 6-well dish, ~90% confluent) were induced with 0.1 µg/mL doxycycline (final) for ~16 hr, solubilised with 500 µL of 1xNLB (1X PBS, 0.3M NaCl, 1% Triton X-100, 0.1% TWEEN 20) + 1x PhosSTOP, sonicated with Bioruptor (30 s ON/OFF, five cycles on LO) and centrifuged for 10 min at ~20.000 *rcf* at 4°C to remove cellular debris. Biotinylated target proteins were purified with 25 µL (slurry) of MyONE-C1 streptavidin beads (Thermo Fisher Scientific, 65002), pre-washed with 1x NLB + 1x PhosSTOP, for 2 hr in the cold-room with end-to-end rotation. Beads were washed 3 times with 500 µL of 1x NLB (5 min each), bound proteins were eluted with 50 µL of 1xLDS sample buffer (Thermo Fisher Scientific, NP0007) + 100 mM beta-mercaptoethanol at 95°C for 5 min. Eluates were loaded on a 4–12% Bis-Tris gel (Thermo Fisher Scientific, NP0322PK2) and transferred to a 0.45 µm PVDF membrane (Merck Millipore, IPVH00010) with 10 mM CAPS (pH 11) + 10% MeOH, for 900 min at 20V. Primary antibodies were used at a dilution of 1:1000 in SuperBlock (Thermo Fisher Scientific, 37515). Membranes were incubated with the diluted primaries overnight in the cold-room.

SC35 and IgG immunoprecipitations (*Figure 1C*) were carried out using a whole-cell extract prepared from wild-type HEK293 cells. Briefly, ~10 million cells were resuspended with 600 µL of 1x NLB + 1x cOmplete Protease Inhibitor Cocktail + 1x PhosSTOP, and kept on ice for 15 min. The lysate was cleared by centrifugation at ~20.000 *rcf* for 10 min at 4°C. Clarified lysate was split into two tubes; to one tube 25 µL of control IgG1 (Santa Cruz, sc-3877) was added, to the other 2.5 µL of SC35 mAb (Sigma-Aldrich, S4045), immune-complexes are allowed to form for 3 hr in the cold-room with end-to-end rotation. 40 µL of Protein G Dynabeads (Thermo Fisher Scientific, 10003D, washed and resuspended with 200 µL of 1x NLB + PI + PS) was used to pull-down target proteins. Beads were washed three times with 1xNLB, briefly with HSB (50 mM Tris.Cl pH 7.4, 1M NaCl, 1% IGEPAL CA-630, 0.1% SDS, 1 mM EDTA) and finally with NDB (50 mM Tris.Cl pH 7.4, 0.1M NaCl, 0.1% TWEEN 20). Bound proteins were eluted with 50 µL of 1xLDS sample buffer (Thermo Fisher Scientific, NP0007) + 100 mM beta-mercaptoethanol at 80°C for 10 min. Immunoblotting was carried out as described for streptavidin pull-downs, except transfer was carried out with 25 mM Tris, 192 mM glycine, 20% (v/v) for 90 min at 90V in the cold-room.

Pull-down of truncated SRRM2 proteins (*Figure 2C–E*) were carried out using whole-cell lysate prepared from respective HAP1 cell lines. The protocol is essentially identical to SC35 and IgG immunoprecipitations described above, with these notable differences: (1) For pull-downs, 25 µL (slurry) of GFP-trap agarose beads were used (Chromotek, gta), incubations were carried out overnight in the cold-room (2) For *Figure 2C* and *Figures 2D*, 3-8% Tris-Acetate gels (Thermo Fisher Scientific, EA0375PK2) were used, for *Figure 2E* a 4–12% Bis-Tris gel was used (3) Gels were run at 80V for 3 hr (4) Transfers were carried out with 10 mM CAPS (pH 11) + 10% MeOH for 900 min at 20V.

## Antibodies

ADAR1(Santa Cruz Biotechnology, sc-73408), alpha-Tubulin (Santa Cruz Biotechnology, sc-32293), COIL (Cell Signaling Technology, D2L3J, #14168), DHX9 (Abcam, ab183731), FLAG-M2 (Sigma, F3165), GFP (Chromotek, PAGB1), Myc-tag (Cell Signaling Technology, 9B11, #2276), Phospho-Rpb1 CTD (Ser2) (Cell Signaling Technology, E1Z3G, #13499), PNN (Abcam, ab244250), RBM25 (Sigma, HPA070713-100UL), SC-35 (Sigma, S4045), SRSF1 (SF2/ASF, Santa Cruz Biotechnology, sc-33652), SON (polyclonal rabbit, Sigma, HPA023535), SRRM1 (Abcam, ab221061), SRRM2 (Thermo Fisher Scientific, PA5-66827), SRSF2 (Abcam, ab28428), SRSF7 (MBL, RN079PW), U1-70K (Santa Cruz Biotechnology, sc-390899), U2AF65 (Santa Cruz Biotechnology, sc-53942).

## Note on SC35/SRSF2 antibodies

There are many commercially available antibodies that are labeled as 'SC35', however only some of them are actually clones of the original SC-35 antibody reported by Fu and Maniatis in 1990. These are: s4045 from Sigma-Aldrich, sc-53518 from Santa Cruz Biotechnology and ab11826 from Abcam. Some antibodies are sold as 'SC35' antibodies, but they are antibodies specifically raised against SRSF2. These are: ab204916 and ab28428 from Abcam and 04–1550 from Merck (can be found with the clone number 1SC-4F11). Neither list is exhaustive.

## Phylogenetic analysis

Unless indicated otherwise, all data analysis tasks were performed using Python 3.7 with scientific libraries Biopython (*Cock et al., 2009*), pandas (*McKinney, 2010*), NumPy (*van der Walt et al., 2011*), matplotlib (*Hunter, 2007*) and seaborn. Code in the form of Jupyter Notebooks is available in GitHub repository: https://github.molgen.mpg.de/malszycki/SON_SRRM2_speckles.

Vertebrate SRRM2, SON, PRPF8, SRRM1, RBM25, Pinin, and Coilin orthologous protein datasets were downloaded from NCBI's orthologs and supplemented with orthologues predicted for invertebrate species. For this purpose, OrthoFinder (*Emms and Kelly, 2019*) was used on a set of Uniprot Reference Proteomes. Invertebrate orthologues were then mapped to NCBI RefSeq to remove fragmentary and redundant sequences. The resulting dataset can be accessed here: https://doi.org/10.5281/zenodo.4065244 and was manually curated to remove evident artefacts lacking conserved domains or displaying striking differences from closely related sequences. Protein lengths were plotted using the seaborn package and descriptive statistics calculated using the pandas package.

In order to resolve phylogenetic relationships between species contained in SRRM2 and SON datasets, organism names were mapped to the TimeTree (*Kumar et al., 2017*) database. Disorder probability was predicted using IUPred2A (*Mészáros et al., 2018*) and MobiDB-Lite (*Necci et al., 2017*) and plotted as a heatmap using matplotlib.

## Acknowledgements

We thank Florian Heyd, Denes Hnisz and Alexander Meissner for critical reading of the manuscript and helpful suggestions. We thank the Mass Spectrometry Facility at MPI-MG, especially Beata Lukaszewsa-McGreal, members of the Microscopy and Cryo-electron Microscopy facility at MPI-MG, especially Thorsten Mielke, Beatrix Fauler and René Buschow. We thank Christina Riemenschneider (Laboratory of Alexander Meissner) and Mirjam Arnold (Laboratory of Andreas Mayer) for sharing reagents, Prof. Feng Zhang and Prof. Veit Hornung for sharing their plasmids via Addgene, developers and maintainers of Open-source software, especially ilastik, CellProfiler, STRINGdb, Python, SciPy, NumPy, Matplotlib, Jupyter Lab and seaborn. Initial cell culture work was only possible thanks to the help of Sebastiaan Meijsing Group. Research in the laboratory of TA is funded by the Max-Planck Research Group Leader program.

## Additional information

### Funding

| Funder | Grant reference number | Author |
|---|---|---|
| Max-Planck-Gesellschaft | Max Planck Research Group Leader Program | Tuğçe Aktaş |

The funders had no role in study design, data collection and interpretation, or the decision to submit the work for publication.

### Author contributions

İbrahim Avşar Ilik, Conceptualization, Data curation, Software, Formal analysis, Supervision, Validation, Investigation, Visualization, Methodology, Writing - original draft; Michal Malszycki, Investigation, Visualization; Anna Katharina Lübke, Claudia Schade, David Meierhofer, Investigation; Tuğçe Aktaş, Conceptualization, Supervision, Writing - original draft, Project administration

Author ORCIDs
İbrahim Avşar Ilik (ID) https://orcid.org/0000-0003-0966-9504
David Meierhofer (ID) http://orcid.org/0000-0002-0170-868X
Tuğçe Aktaş (ID) https://orcid.org/0000-0003-1599-9454

Decision letter and Author response
Decision letter https://doi.org/10.7554/eLife.60579.sa1
Author response https://doi.org/10.7554/eLife.60579.sa2

## Additional files

### Supplementary files
• Supplementary file 1. The oligos used for: 1. the cloning of SRSF1-12 constructs for making stable-cell lines (shown in *Figure 1* and *Figure 1—figure supplement 1*), 2. guide RNAs used to generate SRRM2 truncation HAP1 cell lines and 3. the genotyping oligos used for characterization of SRRM2 truncation HAP1 cell lines (shown in *Figure 2* and *Figure 2—figure supplement 1*).

• Supplementary file 2. The MaxQuant processed output files showing peptide and protein identification, accession numbers, % sequence coverage of the protein, and q-values. (shown in *Figure 1* and *Figure 1—figure supplement 1*) are listed.

• Transparent reporting form

### Data availability
All data generated or analysed during this study are included in the manuscript and Supplementary files and source data files. Mass Spectrometry results shown in Figure 1 and Figure1- figure supplement 1 are provided in Supplementary File 2. Furthermore, they are on ProteomeXchange with identifier PXD021814. Source data files have been provided for Figure 4, Figure 4 - figure supplement1 and 2; and Figures 5, Figure 5 - figure supplement 3 and 4. These zipped files contain ilastik models, CellProfiler pipelines, results and Jupyter notebooks.

The following datasets were generated:

| Author(s) | Year | Dataset title | Dataset URL | Database and Identifier |
|---|---|---|---|---|
| Malszycki M, Ilik IA | 2020 | Orthology and disorder prediction of metazoan SRRM2, SRRM1, SON, Coilin, Pinin, PRPF8 and RBM25 | https://doi.org/10.5281/zenodo.4065244 | Zenodo, 10.5281/zenodo.4065244 |
| Ilik İA, Malszycki M, Lübke AK, Schade C, Meierhofer D, Aktaş T | 2020 | SON and SRRM2 form nuclear speckles in human cells | http://proteomecentral.proteomexchange.org/cgi/GetDataset?ID=PXD021814 | ProteomeXchange, PXD021814 |

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
