## [Decision Letter]

**Acceptance summary:**

Overall, this study provides surprising and important insight into a commonly used mAb and valuable new perspectives on nuclear speckles, perspectives that have the potential to transform future studies. The study will be of broad interest to those interested in splicing, nuclear speckles, antibody specificity, and more generally, liquid-liquid phase separation.

**Decision letter after peer review:**

Thank you for submitting your article "SON and SRRM2 form nuclear speckles in human cells" for consideration by *eLife*. Your article has been reviewed by three peer reviewers, including Jonathan P Staley as the Reviewing Editor and Reviewer #1, and the evaluation has been overseen by Kevin Struhl as the Senior Editor.

The reviewers have discussed the reviews with one another and the Reviewing Editor has drafted this decision to help you prepare a revised submission.

Summary:

This study has yielded two significant contributions. First, the study recharacterized a widely used antibody, mAb SC35, which was initially raised against the spliceosome and characterized both as targeting the 35 kDa protein, SRSF2, an intensely studied splicing regulatory factor, and as marking nuclear speckles, which in the last several years have attracted significant attention for their association with transcriptionally active chromosome regions (after largely being ignored by most for the previous 20 years). The authors present a series of rigorously designed and carefully carried out experiments demonstrating that the 35 kDa factor that mAb recognizes is instead SRSF7. Moreover, the authors present compelling evidence that the primary target of mAb SC35 is a ~300 kDa protein, SRRM2, a spliceosomal factor originally discovered as a nuclear matrix factor and later defined as a nuclear speckle component. In the most convincing experiments establishing these targets the authors show that mAb SC35 signals shift, when the molecular weight of SRSF7 or SRRM2 is varied, and that the signal disappears when SRSF7 is depleted. Given the use of mAb SC35 for nearly three decades, these results suggest that tens if not hundreds of papers require re-interpretation. This study reminds us again the necessity of rigorous validation of antibodies.

Second, the authors investigate the role of SRRM2 in the formation of nuclear speckles. Previous studies have shown that knock down of the nuclear speckle factor SON leads to a compaction of nuclear speckles but not their entire dissolution, implicating a role for at least one additional factor in nuclear speckle formation; other studies have implicated an array of factors as being required for nuclear speckle formation. Here, the authors show that truncation or knock down of SRRM2, in contrast to several other nuclear speckles factors, also reduce nuclear speckle number, although more modestly than SON, and the truncation or knockdown of SRRM2 in combination with the depletion of SON reduces nuclear speckles more than SON depletion alone. The authors interpret these findings to indicate that SON and SRRM2, both of which harbor intrinsically-disordered domains, form nuclear speckles in human cells, as the title indicates. Further, the authors suggest that the double knockdown provides a new tool to study nuclear speckle function. Overall, this study provides surprising and important insight into a commonly used mAb and valuable new perspectives on nuclear speckles, which have the potential to transform future studies. The study will be of broad interest to those interested in splicing, nuclear speckles, antibody specificity, and more generally, liquid-liquid phase separation. We recommend this paper for publication, once the comments are addressed.

Essential revisions:

1) Regarding the significance of SON and SRRM2 to nuclear speckles, the authors need to address the following concerns:

a) The title and the conclusion that SON and SRRM2 form nuclear speckles are not supported by the data. The data show that SON and SRRM2 are necessary for nuclear speckle formation. They do not rule out that another factor is necessary, such as SRRM1, which interacts with SRRM2 and itself harbors an intrinsically-disordered domain; just one protein that is not bound by SON or SRRM2 but still stains nuclear speckles after the double KD would be inconsistent with their hypothesis. The authors have not shown that SON and SRRM2 are also sufficient for nuclear speckle formation. In the absence of new data, the authors need to moderate their conclusions (and title) accordingly.

b) The Abstract implies that the demonstration of SON as a central component of speckles is new ("elusive core"). As appropriately referenced in the text, this is not the case, rather SON is often used as a marker for nuclear speckles, and SON has long been considered to be part of the core of speckles, as knock-down has been documented by several groups to disrupt speckles. The wording in the Abstract should therefore be more parsimonious.

2) The kinase manipulations, discussed in this paper, to eliminate nuclear speckles for experimental purposes are concerning, because such cells appear very sick after these manipulations. Our own experience indicates that the cells after a double, SON/SRRM2 KD also are not very normal. If the authors suggest the SON and SRRM2 double KD as an experimental tool to disrupt nuclear speckles in order to access nuclear speckle function, then it is essential that they indicate cell toxicity, etc. Many SR-protein KDs for example do not allow selection of stable cells. What about this double KD?

3) In principle, in the immunofluorescence studies, the disappearance of mAb SC35 signal on depletion of SRRM2 does not alone prove that SRRM2 is what is visualized by the mAb SC35 in such assays. Given that this paper seeks to establish rigorously that mAb SC35 marks nuclear speckles by recognition of SRRM2, given that SRSF7 is recognized by the antibody on blots, and given that SRSF2 has been traditionally presumed the target of mAb SC35 in nuclear speckles, the rigor of this study demands that SRFS7 and SRSF2 be visualized in cells in the presence of an SRRM2 truncation to rule out that either SRSF7 or SRSF2 phenocopy SRRM2 in this assay. Alternatively, the authors need to recognize this possibility as a caveat to their conclusions.

4) Since the 1990s it has been widely known that the SC-35 mAb has very limited specificity for denatured proteins and was not suitable for immunoblots (see abcam page for ab11826). Indeed, the assumption has always been that it recognizes a folded epitope. Therefore, the use of western blots to conclude anything about the specificity of this antibody is questionable at best, inappropriate at worst. The authors need to address this concern and adjust the manuscript accordingly.

---

## [Author Response]

Essential revisions:1) Regarding the significance of SON and SRRM2 to nuclear speckles, the authors need to address the following concerns:a) The title and the conclusion that SON and SRRM2 form nuclear speckles are not supported by the data. The data show that SON and SRRM2 are necessary for nuclear speckle formation. They do not rule out that another factor is necessary, such as SRRM1, which interacts with SRRM2 and itself harbors an intrinsically-disordered domain; just one protein that is not bound by SON or SRRM2 but still stains nuclear speckles after the double KD would be inconsistent with their hypothesis. The authors have not shown that SON and SRRM2 are also sufficient for nuclear speckle formation. In the absence of new data, the authors need to moderate their conclusions (and title) accordingly.

This is a valid concern and we have thought of the same principal that is if any strongly speckle-associated intrinsically disordered domain containing protein, such as SRRM1 or RBM25, two proteins that are also frequently used as NS markers, would have a similar impact on NS formation as SRRM2 has. To this end, we performed a co-depletion of SON and SRRM1 (shown in Figure 5—figure supplement 5) in a cell line that has a TagGFP2 inserted into SRRM2 gene locus. As it can be seen from the imaging presented in this figure for 4 individual cells (but also more generally on 10 independent field imaged, (data not shown)) we did not score a reduction in the GFP intensity, or dissolution of the spherical bodies as is the case in SON-SRRM2 co-depleted cells. We observed the nuclear speckles have the round-up morphology, that is seen upon SON-KD, but are not dissolved shown with PNN staining and SRRM2-TagGFP signals. Moreover, we performed a co-depletion of RBM25 (another strongly NS-associated protein also used as a NS-marker) and SON which did not result in the dissolution of nuclear speckles (Figure 5—figure supplement 5). Therefore, we have reached to the conclusion that SON and SRRM2 form nuclear speckles with the contribution of SON being more important for the formation and titled our study accordingly.

However, as the reviewer has pointed out, we have not shown that speckles can be reformed by introducing ectopically expressed SON/SRRM2 into cells which now appear not to have nuclear speckles. This would indeed be the formal proof showing that SON/SRRM2 are not just necessary but also sufficient to form nuclear speckles. Such an experiment is quite challenging due to the length of these proteins and difficulty in establishing conditions where one can express these proteins, but not overexpress them which leads to round-up speckles (as shown and discussed by Belmont lab). Therefore, we will change the title to “SON and SRRM2 are essential for nuclear speckle formation” to better reflect our conclusions.

b) The Abstract implies that the demonstration of SON as a central component of speckles is new ("elusive core"). As appropriately referenced in the text, this is not the case, rather SON is often used as a marker for nuclear speckles, and SON has long been considered to be part of the core of speckles, as knock-down has been documented by several groups to disrupt speckles. The wording in the Abstract should therefore be more parsimonious.

We really did try to be clear and just about the previous literature around SON. Indeed, it is clear that SON is a crucial part of NS, likely the most important component for the integrity of speckles. However, in all of these previous studies, RNAi-mediated depletion of SON, without exception, leaves behind spherical bodies that are strongly stained with mAb SC35, that also harbor other NS-markers (which we also show). This is of course not new, as we also appropriately cited previous work, however being able to dissolve these “left-over” speckles by co-depletion of SRRM2, and perhaps more importantly by deletion of the SRRM2’s C-terminal region is indeed novel.

In essence, our results show that in the absence of SON, as shown by previous work as well, NS-associated proteins are still able to organize themselves into nuclear bodies, indicating that either all other SR-proteins without the need of another organizer clump together, or another factor (or factors) is still acting as an organizer. When we remove the C-terminus of SRRM2, which we show is the primary target of SC-35, which strongly stains these left-over nuclear bodies in the absence of SON, then deplete SON, all NS markers that we could find become diffuse, indicating that nuclear speckles no longer exist, or become too small to be detected or classified as “nuclear bodies”. Co-depletion of SON and SRRM2 leads to the same phenotype, but co-depletion of SON and SRRM1 (or RBM25) doesn’t, leaving behind spherical nuclear speckles that harbor SRRM2 which are no different than SON KD cells.

We modified the Abstract and removed the word “elusive” in consideration of this comment.

2) The kinase manipulations, discussed in this paper, to eliminate nuclear speckles for experimental purposes are concerning, because such cells appear very sick after these manipulations. Our own experience indicates that the cells after a double, SON/SRRM2 KD also are not very normal. If the authors suggest the SON and SRRM2 double KD as an experimental tool to disrupt nuclear speckles in order to access nuclear speckle function, then it is essential that they indicate cell toxicity, etc. Many SR-protein KDs for example do not allow selection of stable cells. What about this double KD?

We have stated that our work identifying SON and SRRM2 as the elusive core of nuclear speckles paves the way to study the nuclear speckles under physiological conditions. Here, we have used the cells 24 hours after transfection (~18 hours of knock-down) as the primary reason being that SON-KD caused a mitotic arrest if the cells were kept longer in culture. This was reported earlier in Sharma et al., 2010. There was no additional severity in the phenotype when the SON-KD was combined with SRRM2-KD, therefore we believe the arrest phenotype we scored is mainly due to depletion SON. In this sense, double-depletion of SON and SRRM2 can be used to study the effects of not having NS (transcription, post-transcriptional, topological), but certainly within a time-frame of around 24 hours in cells that haven’t gone through mitosis. We added a clarifying statement at the end of the new “Discussion” section to avoid any misunderstanding as pointed by the reviewer. Faster depletion strategies, and/or a system where cells are mitotically arrested would be required to observe long term effects more reliably.

3) In principle, in the immunofluorescence studies, the disappearance of mAb SC35 signal on depletion of SRRM2 does not alone prove that SRRM2 is what is visualized by the mAb SC35 in such assays. Given that this paper seeks to establish rigorously that mAb SC35 marks nuclear speckles by recognition of SRRM2, given that SRSF7 is recognized by the antibody on blots, and given that SRSF2 has been traditionally presumed the target of mAb SC35 in nuclear speckles, the rigor of this study demands that SRFS7 and SRSF2 be visualized in cells in the presence of an SRRM2 truncation to rule out that either SRSF7 or SRSF2 phenocopy SRRM2 in this assay. Alternatively, the authors need to recognize this possibility as a caveat to their conclusions.

Traditionally, because of the Fu and Maniatis 1992 paper, as pointed out by the reviewer, it is assumed that SC-35 recognizes SRSF2 in immunofluorescence experiments and potentially multiple SR-proteins in immunoblots. The former point, to the best of our knowledge, has never really been proven in any type of rigorous experiment. Fu lab. has generated SRSF2 K/O mice, but never provided an immunofluorescence image that shows that SC-35 signal disappears in K/O cells.

Just to summarize our line of reasoning here:

1) We do an unbiased IP-MS experiment, which shows that SRRM2 is the top candidate protein, at least an order of magnitude away from any other protein in the dataset by any measure. This strongly suggest that SRRM2 is the primary target of this antibody, although doesn’t prove it due to technical reasons i.e. no input normalization, some proteins produce more ‘mass-specable’ peptides than others, and larger proteins tend to produce more peptides.

2) We carry out a biased screen of 12 SR-proteins and find that SRSF7 is strongly recognized by mAb SC-35

3) We do IP-western blotting experiments, which correct for input and are not affected by relative ‘mass-specable’ peptide issues or protein sizes, which reveal a strong enrichment of SRRM2 (>10% of input), some enrichment for SRSF7 (~2% of input) and no enrichment for SRSF2, SRSF1 or other proteins that we have tested.

4) Since the “35kDa” protein is so engrained with the history of this antibody and our results were most consistent with the idea that this protein is SRSF7 rather than anything else, we insert a degron tag to SRSF7. If the hypothesis is true, then we expect a shift of the SC-35 band, concomitant to the shift in SRSF7, which is indeed the case. This is not proof that SC-35 doesn’t recognize any other protein but it does provide very strong evidence (combined with the other two experiments) that the 35kDa band detected by SC-35 in immunoblots is in fact SRSF7.

5) We then show, by TagGFP2 insertion into the SRRM2 locus, that SC-35 mAb can recognize SRRM2 specifically on immunoblots, and furthermore truncations beyond a certain point completely eliminates this signal. We also show later that siRNA mediated KD of SRRM2 also leads to the elimination of the signal from immunoblots (Figure 5—figure supplement 4).

6) Combining the results so far, we address the issue of immunofluorescence, i.e. which protein or proteins are responsible for this signal. We think there are two possible scenarios that could both be true based on the presented evidence so far:

a) This signal is mainly, if not entirely, originates from SRRM2.

b) The signal is a combination of SRRM2, SRSF7 and/or other SR-proteins that the SC-35 might be cross-reacting.

7) We then take advantage of our cell lines with SRRM2 truncations. These truncated SRRM2 version are not recognized by SC-35 mAb on immunoblots, therefore it is reasonable to suspect that they will not be recognized by SC-35 mAb in immunofluorescence as well.

8) If scenario (b) is correct and nuclear speckles are still intact in these cells (which we show that they are indeed intact, judged by SON, RBM25 and SRRM1 stainings Figure 3A-B), then we would expect either no change in SC-35 signal, or a somewhat reduced signal. We see a complete loss of signal.

9) Being extra careful with this result, we also mix the control cell line and SRRM2-truncated cells and image them side-by-side to address any issues related to imaging settings etc. There is no detectable SC-35 signal in truncated cells.

10) We also show that the 35kDa band is still unchanged in SRRM2 truncated cells (Figure 2E), showing that SRSF7 itself is not affected in these cells.

These results, combined together, show that SC-35 signal in immunofluorescence originates from SRRM2, and any other signal potentially contributed by other proteins are below the detection of immunofluorescence microscopy.

We now also additionally provide an IF image that shows SRSF7 staining in tr0 and tr10 cells, imaged side-by-side, showing no observable change in SRSF7 (Figure 3—figure supplement 1). Surprisingly in the historical context, not so surprisingly with the evidence we provide here, we couldn’t find an SRSF2 antibody that shows a significant enrichment of SRSF2 in nuclear speckles (we tried abcam ab204916 and Millipore # 04-1550). The transgene we created can be seen to weakly localized to nuclear speckles Figure 1—figure supplement 2B, we also created an endogenously TagGFP2-tagged SRSF2 that is also weakly localized to nuclear speckles (data not shown). We also haven’t come across a published work that uses a specific SRSF2 antibody (that is, any antibody but SC-35) and shows that SRSF2 co-localizes with another NS marker. Every study that we could find, including all work from the Fu lab., uses transgenic SRSF2 and show that this ectopically expressed form somewhat localizes to NS.

We do acknowledge two caveats though:

1) Our results are all obtained from human cells, which is a reasonable choice of model since the antibody was raised against a human spliceosomal preparation. However, SC-35 mAb has been used by in non-human cells as well. While our results will likely hold in other mammalian species, as we show in our evolutionary analysis, this may not be the case for other species since SRRM2’s C-terminus, part of which is recognized by SC-35, shows a remarkable variation in length and composition. Therefore, we cannot know what this antibody may or may not stain in these organisms.

2) Nuclear speckles completely break down and then re-assemble during mitosis. This event might transiently hide or reveal epitopes that might be recognized by SC-35. We haven’t addressed this issue in this manuscript.

We therefore added the following statement to clarify these points:

“It is important to note that these results were obtained from unsynchronised human cells, which are mostly at the interphase stage of the cell-cycle. mAb SC35 might recognise additional targets in mitotic cells or cells derived from non-mammalian species.”

4) Since the 1990s it has been widely known that the SC-35 mAb has very limited specificity for denatured proteins and was not suitable for immunoblots (see abcam page for ab11826). Indeed, the assumption has always been that it recognizes a folded epitope. Therefore, the use of western blots to conclude anything about the specificity of this antibody is questionable at best, inappropriate at worst. The authors need to address this concern and adjust the manuscript accordingly.

With all due respect to all previous researchers that have used mAb SC35 and published their results, we think that the specificity issue has probably become unnecessarily convoluted due to the initial inaccurate characterization. abcam’s recommendations highlight the issue in an interesting way. In the old marketing images, abcam shows a single band in a total lysate prepared from HEK293 cells:

https://www.abcam.com/ps/products/11/ab11826/reviews/images/ab11826_49518.jpg.

However, producing such an image, in our experience as we have also reported in the manuscript, is only possible under non-ideal western-blotting conditions i.e. when the transfer is not adequate to reveal proteins with large molecular weights. Intriguingly, a customer (not us) complains about an improper WB result obtained with this antibody (with a 2-star rating):

https://www.abcam.com/sc35-antibody-sc-35-nuclear-speckle-marker-ab11826/reviews/68414?productWallTab=ShowAll

It looks like an unexplainable high-molecular smear without the information that we provide in our manuscript, but in light of it, it’s clear that protein stained here is SRRM2.

In our experience the antibody works perfectly fine for western blotting, and very specifically and robustly reveals SRRM2 at ~300kDa, as long as the immunoblotting conditions are optimized for large proteins. We also show that bulk of the signal around 35kDa originates from SRSF7, however as indicated by the other reviewer’s comments, and also previous research, the antibody probably cross-reacts with other proteins as well with varying degree.

In this sense, the antibody can be used for immunoblotting, but pretty much any result obtained from such an experiment must be verified with an independent antibody or independent methods, which we did in this manuscript.